

# The role of the reef-dune system in coastal protection in Puerto Morelos (Mexico)

Gemma L. Franklin[1,2], Alec Torres-Freyermuth[1,2], Gabriela Medellin[1,2,3], María Eugenia Allende-Arandia[1,2,3], Bernabé Gómez[1], Christian M. Appendini[1,2]

[1]Laboratorio de Ingeniería y Procesos Costeros, Instituto de Ingeniería, Universidad Nacional Autónoma de México, Sisal, México
[2] Laboratorio Nacional de Resiliencia Costera, Laboratorios Nacionales CONACYT, México.
[3]CONACYT-Laboratorio de Ingeniería y Procesos Costeros, Instituto de Ingeniería, Universidad Nacional Autónoma de México, Sisal, México

*Correspondence to*: Gemma. L. Franklin (gfranklin@iingen.unam.mx)

**Abstract.** Reefs and sand dunes are critical morphological features providing natural coastal protection. Reefs dissipate around 90% of the incident wave energy through wave breaking, whereas sand dunes provide the final natural barrier against coastal flooding. The storm impact on coastal areas with these features depends on the relative elevation of the extreme
water levels with respect to the sand dune morphology. However, despite the importance of the barrier reefs and dunes in coastal protection, poor management practices have degraded these ecosystems, increasing their vulnerability to coastal flooding. The present study aims to investigate the role of the reef-dune system in coastal protection under current climatic conditions at Puerto Morelos, located in the Mexican Caribbean Sea. Firstly, a nonlinear non-hydrostatic numerical model (SWASH) is validated with experimental data from a physical model of a fringing reef. The numerical model predicts both
energy transformation and runup statistics as compared with experimental results for two different reef crest geometries conducted in a physical model. Thus, the numerical model is further used to investigate the role of the reef-dune degradation in coastal vulnerability. Wave hindcast information, tidal level, and a measured beach profile of the reef-dune system in Puerto Morelos are employed to predict extreme runup and estimate the storm impact scale for different scenarios. The numerical results show that ecosystem degradation has important implications for coastal protection against storms with
return periods of less than 10 years. This highlights the importance of conservation of the system as a mitigation measure to decrease coastal vulnerability and infrastructure losses in coastal areas in the short to medium term.

## 1 Introduction

Coral reefs protect coastal regions against the natural hazards associated with storm wave events, thereby protecting beaches against processes of erosion. Energy dissipation at the coast is increased by the presence of irregular reef surfaces, which are
important in wave transformation (Lowe et al., 2005). These natural barriers can dissipate up to 97% of the incoming wave energy, with the reef crest alone reducing wave height between 64-76% (Lugo-Fernandez et al., 1998; Ferrario et al., 2014).




This property becomes particularly important considering that approximately 850 million people (one eighth of the world's population) reside within 100 km of a coral reef, with more than 275 million living less than 30 km from reefs, benefiting from the services they provide (Burke et al., 2011).

While coral reefs protect the coasts from wave energy, wave-driven flooding along the coast can still occur under extreme events such as hurricanes. However, coral reefs have been degrading over the last four decades (Alvarez-Filip et al., 2009), as a result of a combination of factors including overfishing, coastal development, contamination and excess of nutrients, as well as degradation by coral bleaching events due to increased temperatures. Eakin (1996) reported erosion rates of 0.19 kg $CaCO3$ for a Panama reef, equivalent to a vertical loss of approximately 6 mm $yr^1$ (Sheppard et al., 2005). Considering that reef degradation reduces the protective characteristics of coral reefs, there is an increase in coastal
vulnerability towards extreme events.

The degradation of coral reefs affects the incident wave climate. Nevertheless, the impact of a storm depends not only on the bathymetry and forcing parameters of the storm but also on the geometry of the coast, particularly its elevation (Sallenger 2000). Sallenger (2000) proposed a scale that categorises storm-induced impacts and the magnitude of net erosion and accretion on barrier islands based on the elevation of extreme water levels relative to the elevation of geomorphic
features. Thus, sand dunes act as natural barriers against coastal flooding by attenuating wave energy and slowing inland water transfer (USACE, 2013). After a storm, the height and recovery of the dune are critical for determining the coast's vulnerability to changes in sea level and storms (Durán and Moore, 2013). Although a storm may cause a dune to erode, it provides a source of sediment into the littoral cell (USACE, 2013). This is not the case when the dune is removed by increased coastal development and excessive exploitation of natural resources, which puts these regions at greater risk from
extreme events.

According to a recent report on the importance of coral reefs and dunes (Secaira-Fajardo et al., 2017), the Caribbean is the region that presents the greatest loss of dune vegetation, reducing the vertical extension of the beach and its natural protection. For the case of Cancun, Quintana Roo (Mexico), since 1984 the beach has been receding by 2 m year[-1] as a result of the effects of hurricanes and coastal development (Silva et al., 2006). Construction on the dunes of the barrier island has
restricted eolic transport, thereby preventing the natural regeneration of the dunes (Silva et al., 2006). On the other hand, heights of 3-4 m have been observed for sand dunes in Puerto Morelos (Ruiz de Alegria-Arzaburu et al., 2013). Mariño et al. (2014) pointed out that during the Category 5 hurricane, Wilma, in 2005 the combined presence of dunes, a coral reef, and sand transported from Cancun during the event protected the coast of Puerto Morelos. This suggests that the coast is less vulnerable to extreme events where the reef-dune system is maintained. Unfortunately, coastal dunes in Mexico are at risk
due to coastal or agricultural development (Jiménez-Orocio et al., 2014). Therefore, an assessment on the implications for coastal protection is required.

While there are a number of studies on the role of coral reef geometry on coastal protection (e.g., Quataert et al., 2015), fewer look at the combined effect of reefs and sand dunes on coastal flooding. This study aims to investigate the role of both reef and dune degradation on the storm impact in Puerto Morelos (Mexico). The outline of the paper is as follows.



Section 2 describes the study area and the data employed in this work. The numerical model validation is presented in Section 3. Then, methods used in this study are described in Section 4, followed by the results (Section 5). A discussion on the coastal vulnerability during a hurricane event is presented in Section 6. Finally, concluding remarks are provided in Section 7.

**2 Site and Data description**

The Puerto Morelos fringing reef lagoon is located in the western Caribbean, approximately 25 km south of Cancun on the northeast coast of the Yucatan Peninsula, Mexico (Fig.1a). The climate in the region is hot and humid with a mean annual air temperature of 26.4°C, a maximum of 34.5°C in the summer and a minimum of 13°C in the winter (Merino and Otero, 1991). Rainfall is present all year round, although more intense during the summer, with a mean annual rainfall of 1,041 mm

(Caribbean Coastal Marine Productivity Program: CARICOMP, unpublished data for the period 1993-1998). Evaporation varies from 102 mm in December to 178 mm in May (Merino and Otero 1983). The mean relative humidity is 84% (CONANP 2000). The water temperature at the bottom of the lagoon varies seasonally by around 5°C, from 31-32°C in August and September, to 24-25°C between December and March (Coronado et al., 2007). This area is of particular interest for several reasons, notably its economic importance for tourism and fisheries (10 fishing cooperatives operate in the area),

and its ecological significance, forming a natural protected area.

Puerto Morelos is characterised by a semi-diurnal microtidal regime with a tidal range of less than 0.4 m (Parra et al., 2015). There is also evidence of a low frequency, energetic oscillation (~ 0.4 m), associated with the Yucatan Current and atmospheric pressure which has a period of ~15 days (Coronado et al., 2007; Parra et al., 2014). The wave climate is dominated by wind waves from the Caribbean (South-southeast, SSE) generated by the trade winds. The waves have an

average annual significant wave height, $H_s$, of 0.8 m and a dominant spectral peak period, $T_p$, between 6 and 8 s (Coronado et al., 2007; Parra et al., 2015). In this region, waves exceeding a height of 2 m are considered high-energy waves, which often occur during the northerlies season, locally known as "Nortes", when anticyclonic cold fronts descend over the Gulf of Mexico into the Caribbean Sea during the winter months (Coronado et al., 2007; Mariño-Tapia et al., 2011; Appendini et al., 2013). Between June and October, tropical cyclones can occasionally generate large waves ($H_s \approx$ 6–12 m; $T_m \approx$ 6–12 s)

(Mariño-Tapia et al., 2008). One example of such a storm was Hurricane Wilma, which made landfall on the 23rd of August, 2005 with $H_s >$12 m and a $T_p$ of 10-12 s (measured at a depth of 20 m) (Silva et al., 2012; Mariño-Tapia et al., 2008).

The coastline in the study area is protected by a fringing reef which forms a relatively shallow lagoon of 3-4 m depth and a width that varies from 550 m to 1,500 m (Coronado et al., 2007). The lagoon bed is characterised by calcareous sand covered by patches of seagrass with occasional coral colonies. The reef has a well-developed back-reef and crest

consisting of relatively shallow, submerged coral banks, which play an important role in dissipating wave energy through an active surf zone, thereby protecting the coast. The gently sloping fore-reef descends to an extensive sand platform at a depth





of 20-25 m. The shelf edge is located at a depth of 40-60 m, followed by a subsequent drop-off at approximately 10 km from the coast to depths exceeding 600 m (Ruíz-Rentería et al., 1998).

The width of the beach is relatively stable, ranging between 85-90 m, with a dune of approximately 4 m in elevation, which has been degraded in many areas as a result of coastal development. The mean grain size of the sediment is

0.3 mm, and the mean lower beach slope is 0.05 (Ruiz de Alegría-Arazaburu et al., 2013). The beach profile used in the present study for Puerto Morelos was measured using a Differential Global Positioning System (DGPS) and was provided by CINVESTAV-Mérida. From the beach profile to a depth of 20 m, the bathymetry obtained from CONABIO (http://www.conabio.gob.mx/informacion/gis/) was used (Fig.1b). Wave information for a depth of approximately 20 m is available from a 30-year hindcast (1979-2008) for the Gulf of Mexico and the Western Caribbean Sea (Appendini et al.,

2014). These data were estimated using a third-generation spectral wave model forced with wind data from the North American Regional Reanalysis (NARR) (Mesinger et al., 2006). The numerical model was validated/calibrated in deep waters with wave buoys and altimeter information (Appendini et al., 2013, 2014). The model performance was found to be satisfactory for the Caribbean Sea with an $r^2$ of 0.87 (Appendini et al., 2014). Thus, this information is employed for the boundary conditions in the model.

**3 Numerical Model Validation**

The Simulating WAves till Shore (SWASH) model is used, which is a phase-resolving nonlinear non-hydrostatic model (http://swash.sourceforge.net) developed at Delft University of Technology (Zijlema et al., 2011). This numerical model solves the nonlinear shallow water equations, including the terms for non-hydrostatic pressure, which make it suitable for simulating wave transformation as a result of nonlinear wave-wave interactions in the surf and swash regions. The model is

also capable of simulating wave-current interaction, wave breaking (Smit et al., 2013; de Bakker et al., 2015), wave transformation on reefs (Torres-Freyermuth et al., 2012; Zijlema et al., 2012), and wave-runup (e.g., Brinkkemper et al., 2013; Ruju et al., 2014; Guimarao et al., 2015; Medellín et al., 2016). For further details, including model equations see (Zijlema et al., 2011).

Laboratory experiments with a physical model of a reef were carried out in the wave flume at the Engineering and

Coastal Processes Laboratory of the Universidad Nacional Autónoma de México (UNAM), Sisal. The simplified reef profile (scale of 1:64) was based on that presented by Demirbilek et al. (2007) but with a reef flat of 2.5 m instead of 4.9 m (Fig. 2c). Therefore, the idealised model consisted of a beach with a 1:12 slope followed by a 2.5-m-wide reef flat and a composite slope reef face. Furthermore, a simplified reef crest (2 cm high, 50 cm wide) was added to the profile in the second set of the experiments to investigate the role of the reef crest on beach runup. Different wave conditions were

modelled, including monochromatic waves and Jonswap spectra with various significant wave heights and periods (5-8 cm and 1.75-2.5 s, respectively). Eight sensors were installed at different cross-shore positions along the profile, and the data from these were used to validate the numerical model employed in the present study. In addition to these sensors, a runup





sensor was also installed on the beach face (Fig. 2c). A more thorough description of the simulated cases can be found in Gómez-Pérez (2016).

The results of the physical model were used to validate the SWASH model, with a domain which was 12 m long with a regular grid resolution of $\Delta x = 0.01$. A bottom friction coefficient (*cf*) 0.014 (Manning) was used, which is equal to

that reported previously for a study involving a fringing reef (Peláez et al., 2017). This value is also similar to that reported by Yao et al. (2014 and 2016) for a numerical study on a fringing reef (0.015). The Manning coefficient was used since it best represents the dynamics of waves in the surf zone, compared to other coefficients (Zijlema et al., 2011). A wave breaking parameter ($\alpha$) of 0.6 was used for all simulations. The numerical model was forced with the free-surface elevation time series measured at the offshore sensor for the profiles with and without the reef crest. Laboratory observations were

employed to estimate the significant wave height, wave setup and swash excursion for the reef flat and the reef crest profiles. Figure 2 shows the cross-shore variations in $H_s$ and wave setup ($\eta$) employed for the model validation corresponding to a JONSWAP spectrum with $\gamma=3.3$, $H_s=8.0$ cm and $T_p=2.5$ s. Model simulations show good agreement with respect to the laboratory data ($r^2$ of 0.87 and 0.85 for the reef flat $H_s$ and $\eta$, respectively and $r^2$ of 0.85 and 0.82 for the reef crest $H_s$ and $\eta$, respectively). This is particularly important, since previous studies have demonstrated the importance of accurately

predicting wave setup in the prediction of coastal hazards, including erosion and inundation due to storms (Sheppard et al., 2005; Vetter et al., 2010; Storlazzi et al., 2011; Baldock et al., 2014; Buckley et al., 2015).

Overall the non-breaking wave height is marginally greater for the reef crest profile for both model and data. This is likely due to enhanced wave reflection from the reef face when the crest is included in the profile. There is a rapid decrease in $H_s$ with both profiles at x=7-8 m, related to depth-induced wave breaking. At the same time, the presence of the reef crest

increases the cross-shore radiation stress gradient and hence the maximum wave setup is larger compared to the reef flat, consistent with prior studies (e.g., Yao et al., 2012). Immediately after wave breaking (*x*=8 m), $H_s$ is slightly greater for the reef flat than the reef crest, and wave height becomes depth-limited over the reef flat, as suggested by Storlazzi et al. (2011). Finally, the simulated swash height time series are compared against laboratory observations. A satisfactory agreement is observed for both cases (Fig. 3). Therefore, this numerical model is suitable for conducting a numerical study on wave

transformation and wave runup in the Puerto Morelos reef lagoon.

## 4 Methods

The methodology used in this study is as follows. Firstly, a subset of wave conditions was selected from the 3-hourly 30-year wave hindcast. Selected wave conditions were propagated along different beach profiles, with the corresponding tidal level, from a depth of 20 m to the shore using the SWASH model. Subsequently, the extreme runup $Ru_{2\%}$ and maximum setup

were calculated from the simulated time series, corresponding to each simulated case, and were further employed to re-construct the 30-year extreme water level hindcast using an interpolation technique. Finally, the storm impact was obtained



for different return periods and different scenarios of reef and dune degradation by coupling the extreme water level and dune elevation.

## 4.1 Simulated cases

A total of 87,664 sea states ($H_s, T_p$ and $\theta$) , one every 3 h, comprise the available 30-year wave hindcast (Appendini et al., 2014). A subset of 600 cases was selected, following the method presented by Camus et. al. (2011b) and applied in Medellin et al. (2016), due to the computational effort involved in simulating the entire data set. This method employs the maximum dissimilarity algorithm (MDA) to obtain a subset of wave conditions representative of a variety of sea states (see references for further details). In the present study, the multivariate data included peak period ($T_p$), significant wave height ($H_s$) and

mean sea level ($Z_m$). The wave parameters were obtained from the wave hindcast and the $Z_m$ time series corresponds to the astronomical tide prediction for the same period and location (www.predmar.cicese.mx). The 600 selected sea states were found to adequately represent the whole sample, and were well distributed throughout the time series of sea level and wave parameters (Fig 4a-c). This is consistent with prior studies (Guanche et al., 2013; Medellín et al., 2016).

In the model runs the dune profile was extended beyond the crest, assuming a continuation of the slope measured in

the profile, to complete the model domain and to enable us to infer the effect of reducing the dune crest values (Fig. 5). The model was run with the original profile, which included the back of the dune, and with the extended dune to test whether this affected the wave statistics and no significant differences were found.

## 4.2 Extreme water level calculation

Waves were propagated from a depth of 20 m using SWASH (Zijlema et al., 2011). The SWASH domain extends from a

water depth of 20 m to the shoreline (a distance of 2 km) with a mesh size of 0.1 m. The numerical model was forced using a JONSWAP spectrum at the offshore boundary with the $H_s$ and $T_p$ corresponding to the 600 selected cases from the 30-year wave hindcast and the corresponding sea level according to the tidal prediction. The initial time step was 0.025 s and simulations were sampled for 2,700 s, after 500 s of spin-up time.

For each sea state propagated in SWASH, the height of the bottom profile at the wet-dry interface was used to

extract the water elevation, η(t), relative to mean sea level (Medellin et. al., 2016). To obtain a continuous time series, this location was tracked as the first grid point where water depth was less than 0.01 m. Extreme runup (R), corresponding to the 2% exceedance value in accordance with Stockdon et al. (2006), was calculated for each run. Furthermore, the maximum wave setup at the shoreline, which is the super-elevation of the mean water level due to waves (Longuet-Higgins and Stewart, 1964), was computed as the mean of the wave runup time series (< η >). Therefore, the extreme water levels $R_{high}$

$=Ru_{2\%} + Z$ and $R_{low} =<η> + Z$ were calculated for each case in accordance with Sallenger (2000) and Stockdon et al. (2007).

The 30-year long time series was reconstructed using the extreme water levels from each of the 600 selected sea states, where the storm surge was not included. The time series of extreme water levels were reconstructed using an





interpolation method based on radial basis functions (RBF). Previous studies have identified this method as one of the most suitable for interpolating multivariate scattered data (Franke, 1982) and it has been used to reconstruct time series of wave parameters in coastal waters (e.g., Camus et al., 2011a; Guanche et al., 2013; Medellin et al., 2016).

The RBF interpolation was carried out using an algorithm developed by Fasshauer (2007), which incorporates an
algorithm proposed by Rippa (1999) to obtain an optimal value for a shape parameter that minimizes the interpolation error and is therefore important for the accuracy of the RBF method. This method minimises the root mean square error (RMSE) of a data fit based on a radial interpolant where one of the centres is left out. The RBF was subsequently used to reconstruct the $R_{high}$ time series for the 30 years for all bathymetric profiles studied (see blue line in Fig. 4d). The same method was used to reconstruct the time series of the wave-induced water level, $R_{low}$.

The 30-year reconstructed time series of $R_{high}$ (Fig. 4d) and $R_{low}$ (not shown) were used to assess beach vulnerability under current beach profile conditions (Fig. 5). The return period for both the 30-year $R_{high}$ and $R_{low}$ time series was calculated as the inverse of the probability of a given $R_{high}$ or $R_{low}$ value using the annual maxima data from the reconstructed 30-year time series. Figure 6 shows the return value for $R_{high}$ under the current coastal ecosystem. On the other hand, additional simulations were conducted with beach profiles considering reef degradation scenarios based on 50 yr
projections of reef erosion values reported in the literature (Fig. 5).

**4.3 Storm impact scale for different scenarios**

The storm impact scale proposed by Sallenger (2000) for barrier islands was used to illustrate the implication of changes in reef and beach morphology (reef crest height and dune elevation) on storm-induced water levels. The scale includes four storm impact regimes (Table 1), which depend on the storm-induced water levels and dune elevation, defined as $R_{low}$ (the
astronomical tide, wave setup and storm surge, where included), $R_{high}$ (the sum of the astronomical tide, $R_{2\%}$ and storm surge, where included), $D_{high}$ (dune crest height), and $D_{low}$ (dune toe height). These regimes were calculated for three different reef conditions: (i) present condition, (ii) degraded by 0.30 m, and (iii) degraded by 1.1 m (see Figure 5). For scenario (ii) a vertical loss of 6 mm yr$^{-1}$ was used (reported by Sheppard et al., 2005), while for scenario (iii) the value of 22 mm yr$^{1}$ was used (reported by Eakin 1996). The erosion values reported are a result of El Niño and bleaching events.

**5 Results**

**5.1 Present conditions**

The $R_{high}$ and $R_{low}$ values associated with a 1-, 3-, 5-, 7.5-, 10-, 15- and 30-year return period were then used together with the beach morphology ($D_{high}$ and $D_{low}$), to estimate the storm impact regimes for the present conditions (Table 2). Based on the return values of $R_{high}$ and $R_{low}$, the storm impact regime associated with a yearly return period was "swash" where the
maximum runup is less than the height of the foot of the dune ($R_{high}$>$D_{low}$). For return periods of 3-5 years, the storm impact



regime was "collision" where the maximum runup collides with the foot of the dune but falls below the dune crest ($D_{high} > R_{high} > D_{low}$). For a return period of 7.5 years, the storm impact increases to "overwash" where runup overtops the dune crest and the sand transported landward is lost form the system and does not return to the beach after the storm ($R_{high} > D_{high}$). For return periods of 10 years or greater, the storm impact is "inundation" where the sea level is sufficient that it completely

submerges the dune.

**5.2 Role of reef degradation**

To investigate the role of reef degradation in the reduction of coastal protection we compared the current situation with the scenarios of 0.3 and 1.1 m degradation of the reef crest. It is important to note that in the present study, reef roughness is maintained in all three scenarios to study the effect of the vertical degradation of the reef, although in reality this would

likely be accompanied by a loss of roughness. Figure 7 shows a section of the reconstructed $Ru_{2\%}$ time series for three different reef scenarios (conserved, degraded by 0.3 m and by 1.1 m). Results show a slight increase in $Ru_{2\%}$ when the reef is degraded by 0.3 m, whereas there is a significant increase in $Ru_{2\%}$ when the reef is degraded by 1.1 m. The $R_{high}$ results and the storm impact regimes for the different scenarios support these findings (Fig. 8 and Table 3).

The effect of reef degradation varies depending on the intensity of the storm. For instance, for storms with return

periods of approximately 1-2 years, the increase in $R_{high}$ when the reef is degraded by 1.1 m is almost twofold, whereas the reef degradation of 0.3 m has no visible effect on $R_{high}$ for such return periods (Fig. 8). However, for return periods of 2.5-7.5 years, there is a notable increase in $R_{high}$ for the 0.3 m degraded reef (up to 30%) compared to the conserved scenario (current reef). This is particularly important since most people living on the coast are more likely to experience these storms several times in their lifetimes and relying on the protection provided by the reef will not suffice under a degraded scenario.

For storms with a return period of >10 years the $R_{high}$ values are similar for all three scenarios, this is likely due to the large wave height resulting in the waves breaking further offshore, with the effects of degradation exceeded by the extreme wave conditions. Furthermore, the larger the waves, the more the water depth will increase due to wave setup, making the differences in $R_{high}$ less noticeable because of reef degradation. Also, under certain combinations of energetic wave conditions on fringing reefs, the steep reef face has been shown to facilitate the liberation of fluctuations with infragravity

periods, which can pass into the lagoon with little energy loss and exacerbate the effect of the storm (Roeber et al., 2015). The importance of these long-wave motions inside the lagoon has been previously demonstrated by Van Dongeren et al. (2013). The above phenomenon can be intensified if the reef lagoon resonates with the wave period, amplifying the peak energy of the surf beat (Torres-Freyermuth et al., 2012; Roeber et al., 2015).

Regarding the storm impact regime (Table 2), for a return period of 5 years, there is an increase from a collision

regime to an overwash regime when the reef is degraded 0.3 m. The importance of the reef in protecting the coast becomes more obvious in the scenario where the reef is degraded 1.1 m, showing an increase in storm impact scenarios compared to the other scenarios (conserved and 0.3 m degraded). Based on the results, the degraded 1.1 m scenario will result in the net erosion of the dune even a storm with a yearly return period, whereas inundation will occur for a return period of 7.5 years.





### 5.3 Role of dune degradation

The dune crest elevation is a relevant parameter in coastal protection against extreme water levels. Therefore, the implications of dune degradation can be studied by considering a smaller elevation ($D_{high}$<1.9 m) while estimating the storm impact scale. Model results show that for return periods of 3-10 years the dune degradation by 0.6 m plays a more important

role in coastal protection than the reef crest when degraded 0.3 m (Table 3). Moreover, moderate reef degradation and dune degradation can be more important than the extreme reef degradation of 1.1 m. Therefore, results show the combined importance of conserving the reef-dune system in order to naturally protect the coast from storm conditions. This is similar to the results of Guannel et al. (2016), who found that the greatest nature-based coastal protection is offered when several habitats are considered.

### 6. Discussion: storm impact during hurricane events

Hurricane Wilma (2005) was used to investigate the effect of storm surge on extreme water levels. Thus, a set of runs during the hurricane event were simulated considering only the astronomical tide prediction and a second set including the contribution from the storm surge. We selected wave parameters from the hindcast data between the 19[th] and the 25[th] of October (Figure 9a and 9b), which correspond to Hurricane Wilma, a Category 5 hurricane, which reached the Yucatan

Peninsula on the 20[th]-21[st] of October, 2005. To investigate the storm surge contribution, sea level data were obtained from the HYbrid Coordinate Ocean Model (HYCOM; Halliwell et al., 1998; Bleck, 2001) for the Gulf of Mexico (GoM) (https://hycom.org/data/goml0pt04) for the same dates). For the GoM, HYCOM has a 1/25° or 0.04° equatorial and latitudinal resolution (~3.5 km) for each variable at mid-latitudes. The version of HYCOM used is 2.2.77. Both $H_s$ and $T_p$ from the Hindcast data were interpolated to the same time vector as that of the GoM sea level data (Figure 9).

Figure 9d shows a clear difference in $R_{high}$ when the storm surge is included. The maximum values are higher and the minimum values are lower owing to the storm surge contribution during the hurricane passage. This has implications when calculating the storm regime. In terms of reef degradation and the effects of including the storm surge, the $R_{high}$ values are generally greater for the degraded profiles throughout the five days presented, with the exception of an outlier around the peak of the hurricane, which was an interpolated value, and may therefore not be as accurate as the other values. This may be

because during a hurricane the waves are significantly larger and therefore break further offshore than in non-storm conditions. Therefore, the storm impact during more extreme conditions appears to be less sensitive to reef crest degradation (see Figure 10) than during moderate storm conditions, as noticed previously. It is also important to note that during an extreme event, such as Hurricane Wilma, the reef can act as a barrier against sediment transport, further reducing the storm impact on the coast by retaining sand in the lagoon and on the beach. However, this is not taken into account in the present

study, nor is the effect of changes in reef roughness associated with degradation, which have been shown to have important



implications in wave transformation (Buckley et al., 2015) and wave runup (Osorio et al., 2017). Therefore, the conservation of the dune during such conditions is fundamental for the natural protection of the coastal area.

### 7. Conclusions

A numerical model was employed to study the role of the reef-dune system in coastal protection against extreme wave events in Puerto Morelos (Mexico). The numerical model was validated with laboratory data of wave transformation over a physical model of a reef system. The model-data comparison of free-surface elevation, wave setup, and swash height was satisfactory for simulations with and without a reef crest. Thus, the numerical model was employed to generate a 30-year hindcast of wave-induced extreme water levels for different scenarios of reef-dune degradation in Puerto Morelos (Mexico). The storm impact scale proposed by Sallenger (2000) shows that ecosystem degradation enhances beach vulnerability, particularly for return periods smaller than 10 years. Results show the impact to be more sensitive to dune degradation than crest reduction by 0.3 m, while crest reduction by 1.1 m has a greater effect than degradation of the dune by 0.6 m. The combined degradation of both the dune and reef further increase the vulnerability, so that the conservation of the system as a whole is important for coastal protection. This implies that the environmental service of coastal protection by coral reefs and dunes are critical in the short term regarding infrastructure losses in coastal areas. This becomes particularly important as more people are exposed to sea level rise and coastal hazards (e.g. erosion, flooding, and hurricanes) due to coastal growth (Neumann et al., 2015). The role of reef roughness and two-dimensional processes need to be incorporated for a more comprehensive study.

**Author contribution.** Gemma Franklin carried out all the numerical simulations and data analysis and helped to design the numerical tests. Alec Torres-Freyermuth led and supervised this work and designed both the numerical and laboratory experiments. Gabriela Medellín developed the scripts for downscaling the wave hindcast data and reconstructing the runup time series. María Eugenia Allende-Arandia compiled and helped implement the numerical models in the cluster. Bernabé Gómez carried out all the laboratory experiments. Christian Appendini generated the wave hindcast information employed in this work. Gemma Franklin prepared the manuscript with contribution from all co-authors.

**Acknowledgements.** The first author is funded by a postdoctoral scholarship awarded by the UNAM-DGAPA. Financial support for this study was provided by the National Council of Science and Technology CONACyT through the National Coastal Resilience Laboratory (LANRESC). Alec Torres-Freyermuth, Gabriela Medellín, and Ma. Eugenia Allende-Arandia acknowledge support provided by Cátedras CONACYT project 1146. Many thanks to Gonzalo Martin Ruiz and José López Gonzalez for technical support. Finally, we acknowledge Delft University of Technology for making the development of SWASH possible.





**Competing interests.** The authors declare that they have no conflict of interest.

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



**Table 1. Storm impact scale according to Sallenger (2000).**

| Regime | Description |
| --- | --- |
| Swash | $R_{high} < D_{low}$ |
| Collision | $D_{high} > R_{high} > D_{low}$ |
| Overwash | $R_{high} > D_{high}$ |
| Indundation | $R_{low} > D_{high}$ |





**Table 2. Storm impact regime for the 1-, 3-, 5-, 7.5-, 10-, 15- and 30-year return periods, considering a $D_{high}$ and $D_{low}$ of 1.9 and 1.3 m respectively, for different degrees of reef degradation (0.3 m and 1.1 m).**

| | Storm impact regime | | |
|---|---|---|---|
| Return period | Conserved ($D_{high}$=1.9) | Reef degraded 0.3 m ($D_{high}$=1.9) | Reef degraded 1.1 m ($D_{high}$=1.9) |
| 1 | SWASH ($R_{high} < D_{low}$) | SWASH ($R_{high} < D_{low}$) | COLLISION ($D_{high} > R_{high} > D_{low}$) |
| 3 | COLLISION ($D_{high} > R_{high} > D_{low}$) | COLLISION ($D_{high} > R_{high} > D_{low}$) | OVERWASH ($R_{high} > D_{high}$) |
| 5 | COLLISION ($D_{high} > R_{high} > D_{low}$) | OVERWASH ($R_{high} > D_{high}$) | OVERWASH ($R_{high} > D_{high}$) |
| 7.5 | OVERWASH ($R_{high} > D_{high}$) | OVERWASH ($R_{high} > D_{high}$) | OVERWASH ($R_{high} > D_{high}$) |
| 10 | OVERWASH ($R_{low} > D_{low}$) | OVERWASH ($R_{high} > D_{high}$) | INUNDATION ($R_{low} > D_{low}$) |
| 15 | INUNDATION ($R_{low} > D_{low}$) | INUNDATION ($R_{low} > D_{low}$) | INUNDATION ($R_{low} > D_{low}$) |
| 30 | INUNDATION ($R_{low} > D_{low}$) | INUNDATION ($R_{low} > D_{low}$) | INUNDATION ($R_{low} > D_{Llow}$) |

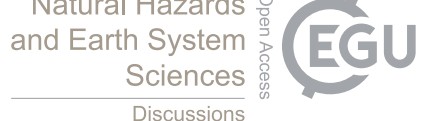

**Table 3. Storm impact regime for the 1-, 3-, 5-, 7.5-, 10-, 15- and 30-year return periods, considering a $D_{high}$ and $D_{low}$ of 1.3 m, for different degrees of dune and reef degradation.**

| Return period | Storm impact regime | |
|---|---|---|
| | Dune degraded reef conserved ($D_{high}$=1.3) | Reef (1.1 m) and dune degraded ($D_{high}$=1.3) |
| 1 | SWASH  ($R_{high} < D_{low}$) | OVERWASH  ($R_{high} > D_{high}$) |
| 3 | OVERWASH  ($R_{high} > D_{high}$) | OVERWASH  ($R_{high} > D_{high}$) |
| 5 | OVERWASH  ($R_{high} > D_{high}$) | OVERWASH  ($R_{high} > D_{high}$) |
| 7.5 | OVERWASH  ($R_{high} > D_{high}$) | OVERWASH  ($R_{high} > D_{high}$) |
| 10 | OVERWASH  ($R_{high} > D_{high}$) | INUNDATION  ($R_{low} > D_{low}$) |
| 15 | INUNDATION  ($R_{low} > D_{low}$) | INUNDATION  ($R_{low} > D_{low}$) |
| 30 | INUNDATION  ($R_{low} > D_{low}$) | INUNDATION  ($R_{low} > D_{low}$) |



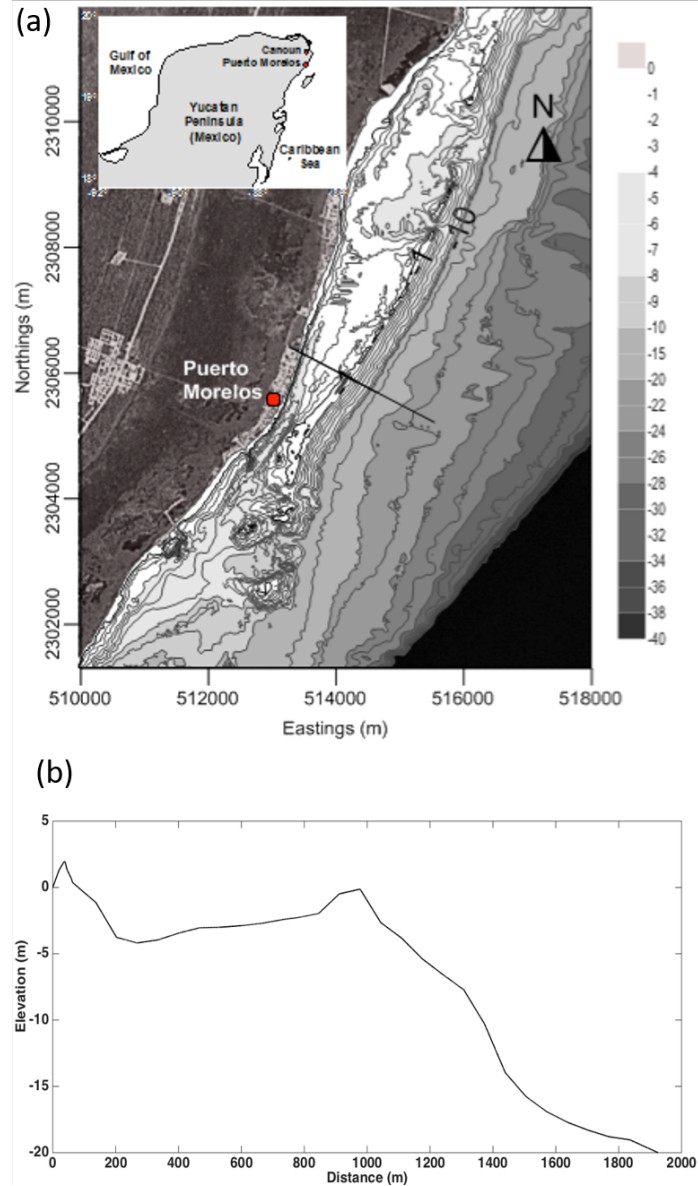

**Figure 1. (a) Map of the study area. The solid black line indicates the location of the bathymetric transect used in the numerical model. (b) Bathymetry obtained from the transect indicated on the map (bathymetry courtesy of CONABIO), including a beach profile surveyed in March, 2014 (courtesy of CINVESTAV-Merida).**




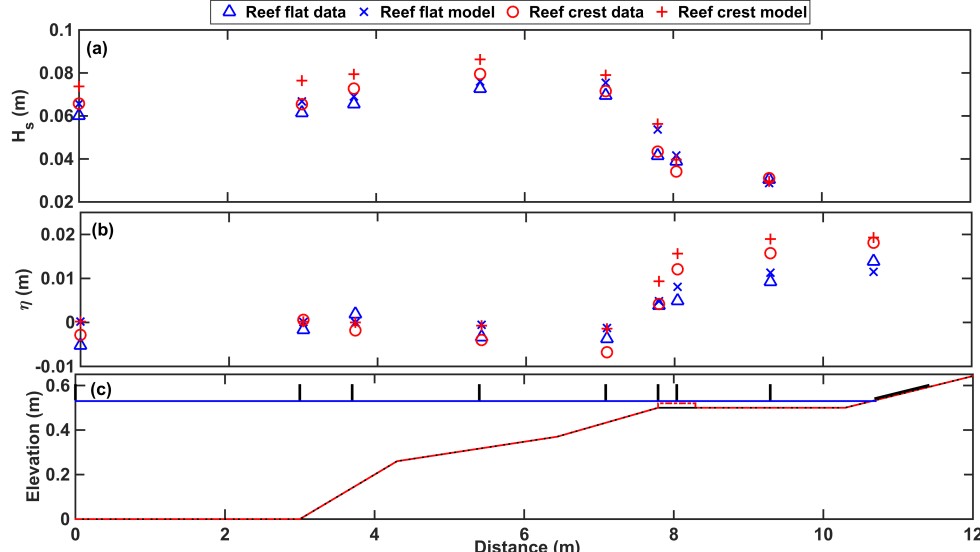

**Figure 2. Model validation using the reef flat and reef crest measurements of (a) significant wave height and (b) wave setup (blue represents the reef flat and red the reef crest). Circles and triangles indicate laboratory data and crosses model data. (c) Profile of the physical model used in the laboratory experiments. The solid black line indicates the reef flat and the dashed red line indicates**
5 **the reef crest profile. The blue line represents the water level used in the experiments, and the solid black lines indicate the locations of the capacitance sensors.**





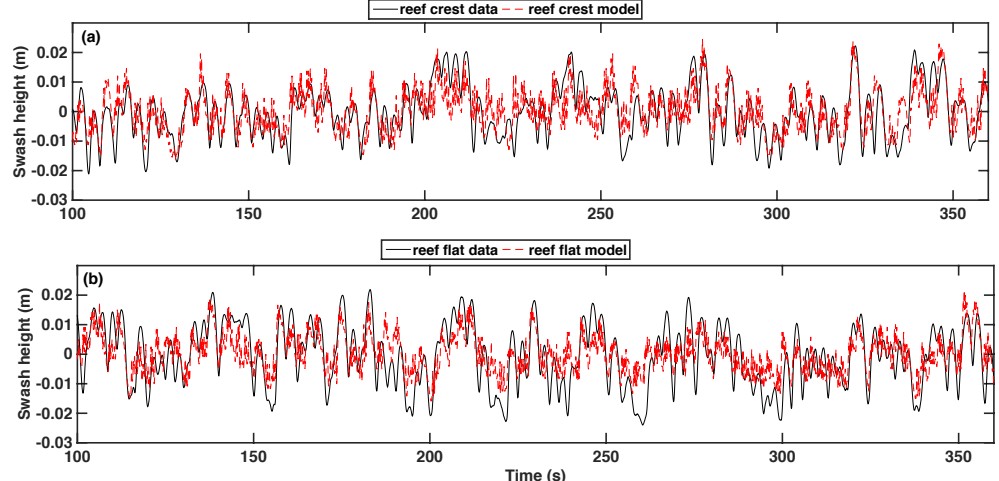

Figure 3. Swash height time series for the reef (a) with and (b) without crest.





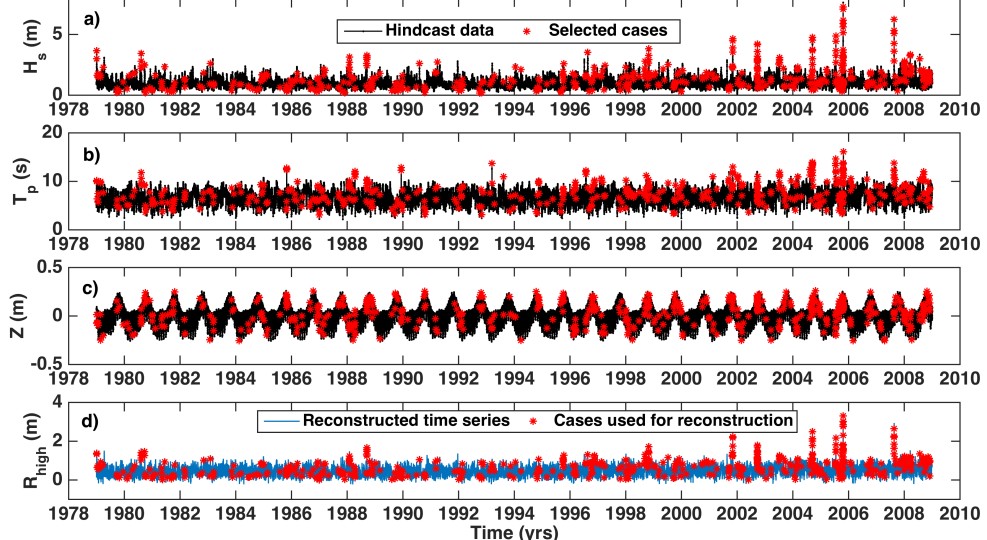

**Figure 4. Reconstructed time series, including R$_{high}$, for the current reef profile. (a)–(c) Black lines indicate available hindcast data and red stars indicate the selected cases used to represent the complete time series. (d) Blue line represents time series reconstructed from the results of the simulated results. Red stars indicate the cases used for reconstruction.**



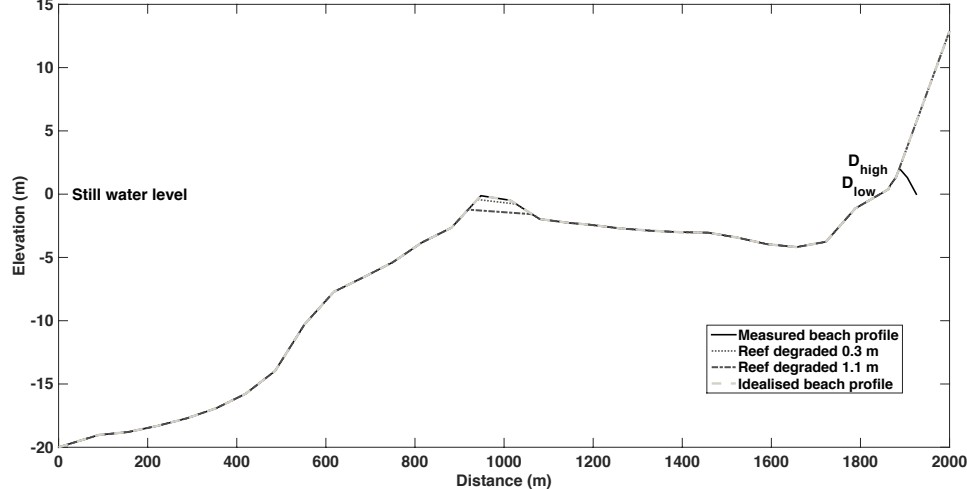

**Figure 5. Measured beach profile (solid black line) and idealised profile with beach extended beyond the dune (dashed grey line). $D_{high}$ represents the dune crest and $D_{low}$ the foot of the dune. The degraded profiles (0.3 m and 1.1 m) are indicated by the dotted dark grey and dashed black lines respectively.**





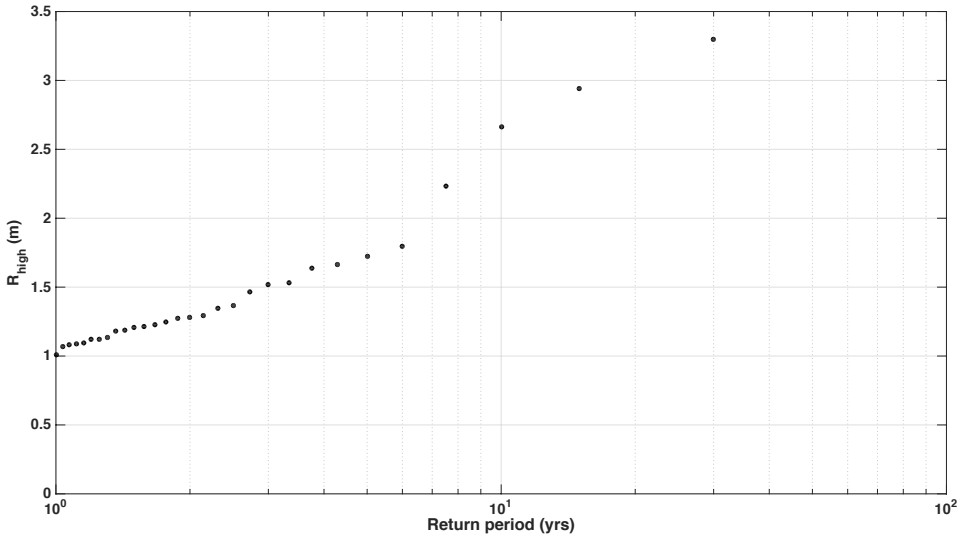

**Figure 6. Return value of R$_{high}$ for the current reef profile.**



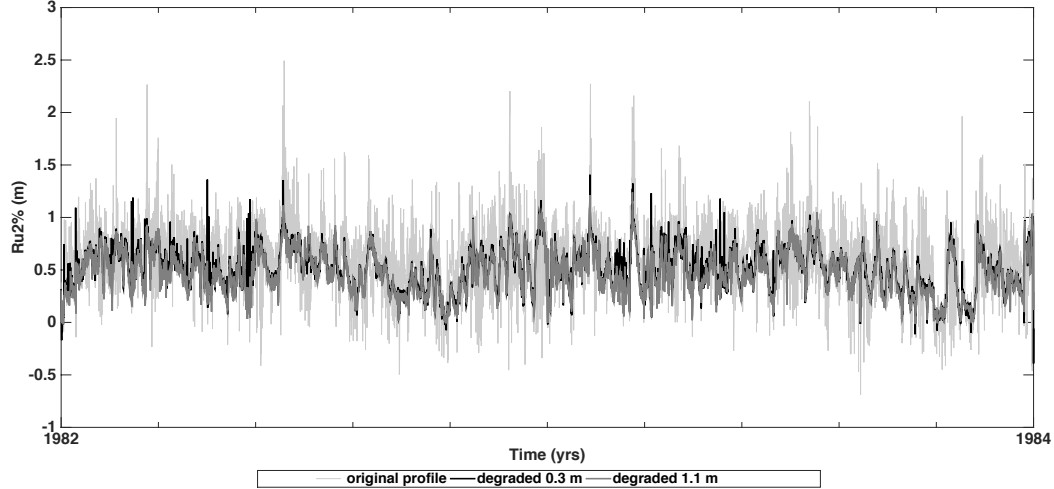

**Figure 7. A section of the Ru2% time series for the different reef profiles.**

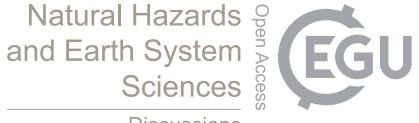



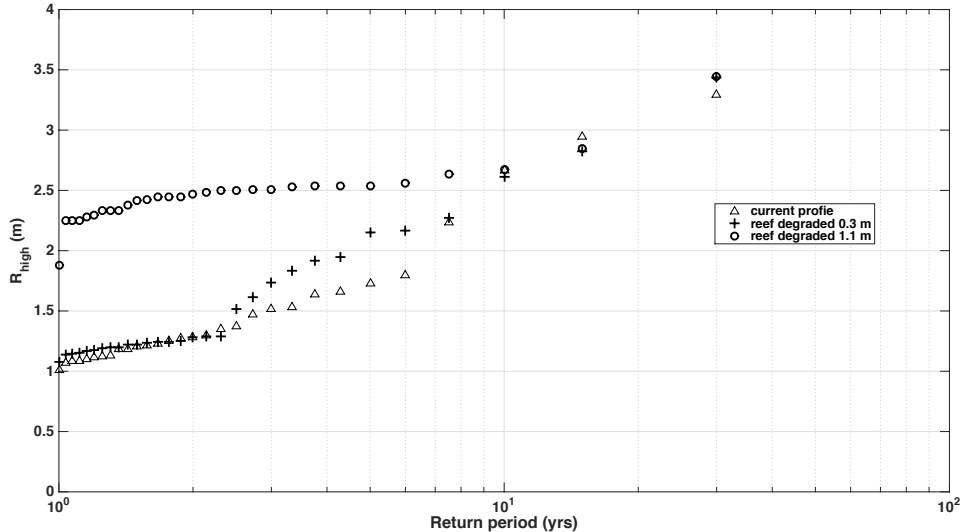

**Figure. 8 Return value of R$_{high}$ for the current reef profile (solid dots), the reef degraded by 0.3 m (crosses) and for the profile with the reef degraded by 1.1 m (open circles).**



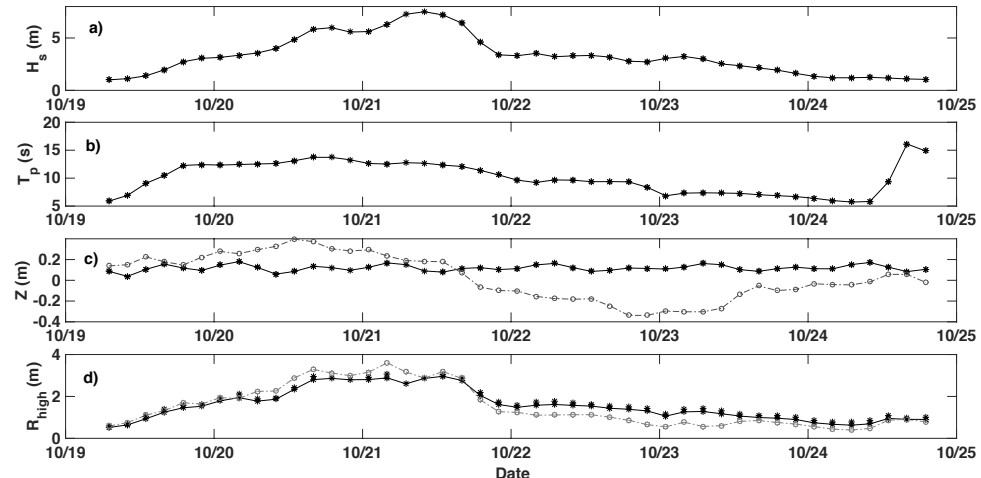

**Figure 9. (a) Significant wave height (*H_s*), (b) peak period (*T_p*), (c) sea level (*Z*) (black: predicted tide, grey: GoM sea level) and (d) R_high (black: without the storm surge, grey: with storm surge) during the pass of Hurricane Wilma (2005).**



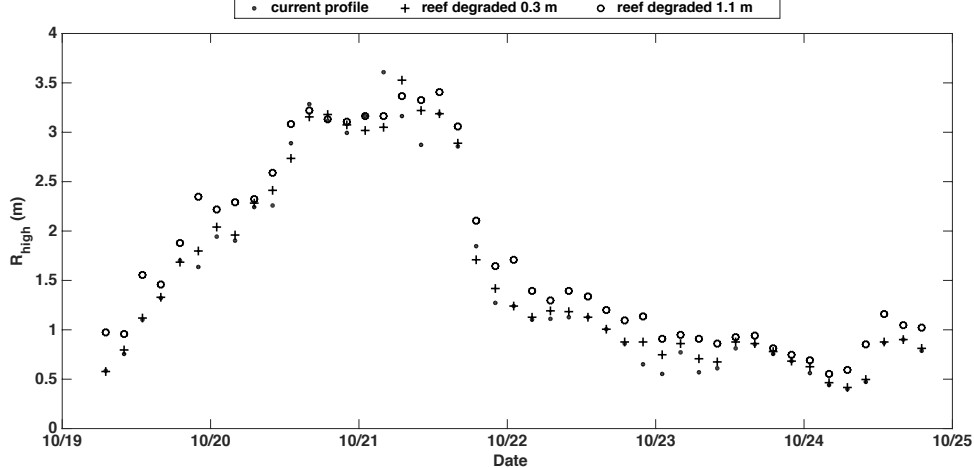

**Figure 10. R_high during Hurricane Wilma (2005) for the current reef profile (solid dots), the reef degraded by 0.3 m (crosses) and for with the reef degraded by 1.1 m (open circles).**