# Peer review of "The role of the reef-dune system in coastal protection in Puerto Morelos (Mexico)"

_Natural Hazards and Earth System Sciences, 2017_

## Referee Comment (RC1) · Anonymous Referee #1 · 4 Oct 2017

Review of "The role of reef-dune systems in coastal protection in Puerto Morelos (Mexico)" by Franklin et al. This paper presents an analysis of the combined impact of reef and dune degradation on determining storm impact. In general I found the paper interesting and conclusions primarily supported (and very timely given recent events), but the analysis a bit lacking. See my detailed comments below, but my general recommendation is that this paper needs a major revision prior to publication. The numerical simulations conducted by the authors can provide much more information about what is causing the observed runup extremes and it would be good to delve a bit deeper into what is going on.

Specific comments 1. Pg 3, lines 5-15. A majority of this information is not relevant, e.g. annual temperature and rainfall do not impact the runup.

[Figure]

2. Pg. 4, line 10. Please specify which model.

3. Pg. 4, line 13. A high r2 does not indicate model performance unless coupled with the regression slope. The r2 only tells you how well a model reproduces the variance.

4. Section 3. I do not see the point in including the flume experiments in this paper. You are essentially calibrating the model on an unrelated data set for a reef/beach profile that was not made to replicate your field site. Essentially you are just showing that SWASH works on reef profiles which has already been shown (Zijlema, 2012, Buckley et al., 2014). Additionally, and while it is unfortunately the case, showing that the model is calibrated at one site does not mean it is calibrated at all other sites. As a result, my preference would be to entirely remove the discussion and comparison with the flume results and use the extra space to further develop the results as they result to the field site. Also I found the discussion of the runs with and without the reef crest confusing.

5. Figure 1. Can the inset be made a higher resolution and zoomed out a bit to provide more geographic context?

6. Pg. 6 line 5. How long is each simulation run for? This is important in determining the validity of the statistics which include long waves.

7. Page 6 line 26. Extreme runup is defined inconsistently.

8. Page 6. Line 30. I am confused about the definition of Rlow. The setup is the average runup so why is this no Ravg? Also it would be helpful to remind the reader that here Z is the tidal level.

9. Page 6 line 32. As is sort of acknowledged in the discussion, not including surge is a huge limitation of the approach. As the depth of reef submergence directly effects the short wave transmission across the reef the surge is critical in determining the runup (in addition to the fact that the surge adds to the water level from which waves runup). Could you not include this for the simulations suing the hycom model? I find this a major limitation of the current study. High surge also acts as a proxy for the reef

degradation, and thus neglecting surge probably causes your results to underestimate the occurrence of over toping.

10. I think the results section could be considerably beefed up. By using a phase resolving model you allow for a lot of information on the runup dynamics to be gleaned. As has been demonstrated in the available literature reef/lagoon systems can often act as open basins and thus have the potential to enhance/trap IG energy.

11. I like the inclusion of the dune height in the analysis but wonder if treating the dune as an un-erodible feature underestimates the overtopping.

---

## Referee Comment (RC2) · Anonymous Referee #2 · 6 Oct 2017

The paper presents numerical modeling and analysis of wave runup on a reef-dune fronted coastline. The material presented is interesting and scientifically relevant. However, the description of methods and analysis of the data are lacking. Variables need to be defined more clearly and the notation (including italics) needs to be consistent throughout the text and figures; wave setup and 2% exceedance runup are two examples of inconsistent notation. The model validation with laboratory data is cursory and doesn't add much to the paper as it is written. The two model validation figures (Figures 2 and 3) do not state what runs are shown and whether it is a monochromatic or irregular wave case. This is extremely important as infragravity waves likely dominate the runup spectra in the field case. I would use the laboratory results and model comparison to highlight some of the important wave and wave setup dynamics, before

going into the field modeling results. Of particular interest would be using the flume/ model results to explain the trends in Figure 8 and 10, which show less difference between reef degradation scenarios under large wave conditions.

Not including storm surge is very problematic to the legitimacy of the hind-cast modeling. I think the authors should either include storm surge or reorganize the paper as more of a theoretical investigation rather than an applied hind-cast analysis. Major revisions are recommended before publication.

See detailed notes below: 1. Figure 1: Label color bar.

2. Figure 2: Need to state the wave forcing and still water level. Also, the measured wave setup is negative offshore due having a fixed volume of water in the flume (Figure 2b). The initial water levels in the model should be adjusted to this offshore water level and rerun with the correct offshore water level.

I would include a low wave and a large wave example to highlight the dynamics seen in Figure 8 and 10.

3. Figure 4: Need to state the wave forcing and still water level. Are Z and Z_m the same? Include definition of Z and R_high in the caption.

4. Figure 6: Is this data repeated in Figure 8? Remove this figure if it is.

5. Figure 8: This figure is extremely interesting. I would reorganize the paper to focus on explaining the trends seen here.

6. Page 2 Line 10: This paragraph needs revision. "The degradation of coral reefs affects the incident wave climate." Are you referring to the offshore wave climate? If so how does coral reef degradation affect offshore conditions?

7. Page 3 Line 5-15: Most of this site description isn't needed.

8. Page 4 Line 30: I would not include monochromatic waves in the model analysis and statistics as infragravity waves will be important for runup in the field.
[Figure]

9. Page 5 Line 5: I don't understand this discussion of bottom friction coefficients. Coefficients used in flume studies are discussed as if they are applicable to field cases? Are you using these values for your hind-cast analysis?

10. Page 5 Line 10: I would either remove the $r^2$ statistics or expand on how these values were calculated and which runs were used with table of runs, etc. I would probably remove and just focus on detailed analysis of two representative wave cases.

11. Page 5 Line 20: Did the flume have active reflection compensation? Were outgoing waves removed from the time series used to force the model? Generally need more details.

12. Page 6 Line 25: In Figure 2 and elsewhere eta is used for wave setup. Here eta is a function of time. Change wave setup to overbar eta or <eta>. Sometimes 2% runup exceedance is written as R sometime Ru2% sometime Ru_2% sometime italics sometimes not. Make this consistent

13. Page 8 Line 20: I would go more into explaining the R_high response at larger wave heights. You mention resonance, but you don't provide evidence or state if this is the cause of the R_high response.

14. Page 10 Line 1: The Buckley et al. 2015 reference should be Buckley et al. 2016

---

## Referee Comment (RC3) · Anonymous Referee #3 · 20 Oct 2017

The manuscript presents a numerical study on the role that a reef-dune system plays in protecting a given coast from storms. The case study of Puerto Morelos, Mexico has been selected to this end. The study illustrates the importance of a holistic management of the coast (considering the reef and dunes as part of a single system) in order to maximise the protective service obtained from ecosystems, which is very relevant in the context of coastal developments, climate change and other factors that compromise the stability of such habitats. Therefore, the paper may potentially be very useful to policy makers, engineers and scientists concerned with a sustainable management of the coast. However, the study also presents some significant weaknesses that should be amended before publication of the manuscript is advised. Please find below a list of points –in decreasing order of importance- that should be addressed by

the authors before I can recommend publication of the present paper in NHESS.

1. A good portion of the manuscript is devoted to the validation of the model (SWASH) against laboratory data, after which the authors conclude that such a validation justifies application of the model to the field case study. The problem with this line of reasoning should be evident and weakens the paper significantly. The numerical model SWASH has previously been validated (extensively) against laboratory experiments, so this section in itself does not add much to the present study. What one would expect instead is a calibration/validation of the model against field data from Puerto Morelos (the site selected for this research) before carrying out the rest of the study. If such data were not available, the manuscript should probably be reformulated as a more theoretical study and all necessary assumptions (e.g. on bed friction coefficients) should be justified.

2. The authors confess (e.g. page 9 line 30) that changes in reef roughness are important, but yet have not been considered in this study. Understandably, some assumptions need to be adopted (such as 1D approach, which may miss many important real 2D phenomena, but is a good first approximation), but variable reef roughness for degraded scenarios does not seem to be particularly cumbersome to include in the simulations. Hence, I would recommend that the authors either include variations in reef roughness for different degradations scenarios or justify why this has not been done.

3. In line with the previous point, study of the effect of a degraded sand dune, by means of a modified dune height, is an interesting aspect of this study. However, I wonder about the validity of the conclusions achieved regarding flooding (storm impact) when the sand dune has been reduced in height but considered non-erodible during the simulation. A discussion on how this assumption affects the conclusions would be valuable. Ideally, inclusion of morphological evolution of the dune/beach profile in the study of protective services provided by the reef-dune system would significantly strengthen the point made by this article (according to the authors themselves; page 9 line 28).
[Figure]

4. The paper could be written in a more concise manner by avoiding excess of uninformative or non-relevant details all throughout the manuscript (especially true for Section 2).

5. No reference is given for the adopted projections of reef erosion (page 7 line 15).

6. I am not sure all figures are very useful or transmit their message in a clear way. For example, Fig 3 could be transformed into a statistical measure of the goodness of fit between model and experiments. Similarly, Fig 7 is not very informative – the y-axis could probably be presented as the percentage increase/decrease in Ru2% with respect to a reference case (e.g. current profile).

7. In general, the manuscript is well structured and written, but is not completely free from typos and grammatically confusing sentences. A general revision of the writing is recommended.

---

## Author Comment (AC1) · 8 Dec 2017

**Response to comments of Referee 1:**
**"The role of the reef-dune system in coastal protection in Puerto Morelos (Mexico)"**

(nhess-2017-304)

by Gemma L. Franklin, Alec Torres-Freyermuth, Gabriela Medellin, María Eugenia Allende-Arandia, Christian M. Appendini

**Referee #1:**
**Review of "The role of reef-dune systems in coastal protection in Puerto Morelos (Mexico)" by Franklin et al. This paper presents an analysis of the combined impact of reef and dune degradation on determining storm impact. In general I found the paper interesting and conclusions primarily supported (and very timely given recent events), but the analysis a bit lacking. See my detailed comments below, but my general recommendation is that this paper needs a major revision prior to publication. The numerical simulations conducted by the authors can provide much more information about what is causing the observed runup extremes and it would be good to delve a bit deeper into what is going on.**

RESPONSE: We thank the referee for his/her comments which have helped us improve the manuscript. A detailed point-by-point response to the referee's comments is provided below. Following the referee's suggestion the revised manuscript includes: (i) a more thorough analysis on the role that storm surge has in the storm impact, and (ii) a detailed study of runup dynamics.

**Specific comments**

**1. Pg 3, lines 5-15. A majority of this information is not relevant, e.g. annual temperature and rainfall do not impact the runup.**

RESPONSE: We agree with the referee and hence most of this information has been removed from this section in the revised manuscript. The following text has been removed from the manuscript:

*"The climate in the region is hot and humid with a mean annual air temperature of 26.4°C, a maximum of 34.5°C in the summer and a minimum of 13°C in the winter (Merino and Otero, 1991). Rainfall is present all year round, although more intense during the summer, with a mean annual rainfall of 1,041 mm (Caribbean Coastal Marine Productivity Program: CARICOMP, unpublished data for the period 1993-1998). Evaporation varies from 102 mm in December to 178 mm in May (Merino and Otero 1983). The mean relative humidity is 84% (CONANP 2000). The water temperature at the bottom of the lagoon varies seasonally by around 5°C, from 31-32°C in August and September, to 24-25°C between December and March (Coronado et al., 2007)."*

**2. Pg. 4, line 10. Please specify which model.**

RESPONSE: The name of the model and reference have been included. The text has been changed to:

*"These data were estimated using the third-generation spectral wave model MIKE 21 SW (Sørensen et al. 2004) forced with wind data from the North American Regional Reanalysis (NARR) (Mesinger et al., 2006)."*

**3. Pg. 4, line 13. A high r2 does not indicate model performance unless coupled with the regression**

**slope. The r2 only tells you how well a model reproduces the variance.**

RESPONSE: Further information on the model performance has been included.

*"…The mean observed wave height ($H_s$) and peak period ($T_p$) were 1.22 m and 6.70 s respectively, compared to the mean reanalysis/hindcast values of 1.31 m and 7.27 s (Rms of 0.33 $H_s$ and 1.59 for $T_p$ with correlation coefficients of 0.90 and 0.51)…"*

**4. Section 3. I do not see the point in including the flume experiments in this paper. You are essentially calibrating the model on an unrelated data set for a reef/beach profile that was not made to replicate your field site. Essentially you are just showing that SWASH works on reef profiles which has already been shown (Zijlema, 2012, Buckley et al., 2014). Additionally, and while it is unfortunately the case, showing that the model is calibrated at one site does not mean it is calibrated at all other sites. As a result, my preference would be to entirely remove the discussion and comparison with the flume results and use the extra space to further develop the results as they result to the field site. Also I found the discussion of the runs with and without the reef crest confusing.**

RESPONSE: The referee's comment is consistent with those from the other two referees in the sense that there is no added value by including the flume experiments in this paper. Therefore, the section on the model validation employing laboratory data has been removed. Therefore, in the revised manuscript we have made reference to previous studies where the model has been validated for reef profiles. The following text has been included in the revised manuscript:

*"The model is also capable of simulating wave-current interaction, wave breaking (Smit et al., 2013; de Bakker et al., 2015), and wave-runup (e.g., Brinkkemper et al., 2013; Ruju et al., 2014; Guimarao et al., 2015; Medellín et al., 2016). Furthermore, previous studies (Torres-Freyermuth et al., 2012; Zijlema et al., 2012; Buckley et al., 2014) have shown good agreement on simulating wave transformation on reef profiles."*

**5. Figure 1. Can the inset be made a higher resolution and zoomed out a bit to provide more geographic context?**

RESPONSE: Following the referee's suggestions Figure 1 has been modified to provide a better geographical context and improve its quality. The revised Figure 1 is shown below.

[Figure]

Figure 1R. (a) Map of the study area in the Gulf of Mexico. The solid black line indicates the location of the bathymetric transect used in the numerical model. (b) Bathymetry obtained from the transect indicated on the map (bathymetry courtesy of CONABIO), including a beach profile surveyed in March, 2014 (courtesy of CINVESTAV-Merida).

**6. How long is each simulation run for? This is important in determining the validity of the statistics which include long waves.**

RESPONSE: The SWASH simulations were run for 3200 s (see line 23, page 6 of the original manuscript), which were sampled for 2700 s after accounting for model spin-up time, considering a constant water level. The simulation time is sufficient to account for long wave statistics.

**7. Page 6 line 26. Extreme runup is defined inconsistently.**

RESPONSE: We thank the referee for pointing out this inconsistency. Extreme value statistics of runup, often defined as the elevation exceeded by only 2% of runup, is denoted by $R_{2\%}$ (Homan, 1986).

**8. Page 6. Line 30. I am confused about the definition of Rlow. The setup is the average runup so why is this no Ravg? Also it would be helpful to remind the reader that here Z is the tidal level.**

RESPONSE: Here, $R_{low}$ is adopted from a previous study (i.e., Sallenger, 2000) where the storm impact scale was introduced for the first time. The term $R_{low}$ consists of the average runup (setup) plus the tidal level and, where applicable, the storm surge. Thus, it represents the low extreme sea level resulting from the aforementioned contributions, which is then compared with morphological features in the beach profile. Therefore, we have maintained this terminology for consistency with previous work (e.g., Stockdon et al., 2007; Medellín et al., 2016; among many others). The revised manuscript includes the following text for further clarification:

*"$R_{low}$ represents the low extreme sea level resulting from the setup, tidal level and storm surge contributions consistent with Sallenger (2000)."*

**9. Page 6 line 32. As is sort of acknowledged in the discussion, not including surge is a huge limitation of the approach. As the depth of reef submergence directly effects the short wave transmission across the reef the surge is critical in determining the runup (in addition to the fact that the surge adds to the water level from which waves runup). Could you not include this for the simulations suing the hycom model? I find this a major limitation of the current study. High surge also acts as a proxy for the reef degradation, and thus neglecting surge probably causes your results to underestimate the occurrence of over toping.**

RESPONSE: We agree with the referee on the importance of including the storm surge. The reason for not including the storm surge contribution in the previous version of the manuscript is the lack of long tidal records in the study area and that the Hycom data only encompasses 16 years of the 30 years of data corresponding to the wave hindcast information. However, we recognise the importance of studying the role of the storm surge using the available information. Therefore, the numerical model has been re-run selecting 300 representative cases, for the 16-year Hycom period (using the same methodology as for the 30 year hindcast), using both the sea surface height obtained from Hycom (including storm surge) and considering only the predicted tide. The numerical results made it possible compare the effect of including this contribution on the extreme water levels and the storm impact. Figure 2R shows $R_{high}$ as a function of the return period while considering the two different scenarios. An increase in $R_{high}$ is observed when storm surge is included. This increase is important since, as recognised by the referee, it acts as a proxy for degradation. When excluded it results in an underestimate of the effects of reef degradation on runup and hence coastal flooding. However, the effect of the storm surge (for the time period available) was smaller than the effect of the reef degrading by 1.1 m but slightly greater than the reef degrading by 0.3 m, particularly for return periods of less than 3 years. These new results are incorporated in the discussion of the revised manuscript.

[Figure]

Figure 2R Return value of $R_{high}$ for the model run with the storm surge (open circles) and without (crosses) for the time period of 1993-2008.

**10. I think the results section could be considerably beefed up. By using a phase resolving model you allow for a lot of information on the runup dynamics to be gleaned. As has been demonstrated in the available literature reef/lagoon systems can often act as open basins and thus have the potential to enhance/trap IG energy.**

RESPONSE: By removing the section on the laboratory validation, there is more room to look into the results in greater detail. Thus, analyses on runup dynamics are further investigated following the referee's suggestion.

Incident and infragravity swash height have been analysed using the parameterisations proposed by Stockdon et al., (2006). For beaches, these authors found incident swash height ($S_{inc}$) to be best parameterised by a dimensional version of an Iribarren–type relationship ($S_{inc}=0.75\beta(H_0L_0)^{1/2}$, where $\beta$ is the beach face slope, $H_0$ and $L_0$ incident wave height and length respectively. Fig. 3Ra shows the incident swash height for the present study (high and low water contributions are presented in green and red respectively). The 15% exceedance value of water level according to the astronomical tide $Z$ was used for high ($Z \geq Z15\%=0.1636$ m) and low water level ($Z \leq Z15\%=-0.1636$ m). As shown in the figure, Stockdon's parameterisation works fairly well for $S_{inc}$, particularly for high water levels, although it slightly over predicts the numerical results. Figure 3Rb shows the results of using the same parameterisation for infragravity swash height ($S_{ig}$), as well as the effect of replacing the beach slope parameter ($\beta$) with the reef face slope ($\beta_{reef}$) (blue vs. cyan line), which results in an improved fit. Stockdon et al. (2006) found that by excluding beach slope in the parameterisation resulted in the best fit for $S_{ig}$ (Fig 3Rc), which also works fairly well for the high water level $S_{ig}$ values for the present study, although less applicable for more energetic waves. A notable difference between the runup contributions on reef-protected beaches with respect to sandy beaches is that $S_{ig}$ contributions were considerably larger. In order to look at this further, $S_{inc}$ vs. $S_{ig}$ variance was plotted against the Iribarren number (Fig. 4R), showing a clear dominance of $S_{ig}$ contributions under practically all wave conditions. This demonstrates a key difference in the swash contributions on beaches compared to reef environments, where infragravity dominates.

With regards to wave setup $\langle\eta\rangle$, the parameterisations presented by Stockdon et al. (2006), with (a) and without (b) beach face slope, underestimate wave setup for a reef environment (Fig. 5R). The effects of the relative contributions of high and low water to wave setup are less obvious for this profile than for sandy beaches (e.g. Medellin et al., 2016). When the slope of the reef face is used instead of the beach face slope, the parameterisation improves (cyan versus blue line Fig. 5Ra), although it still underestimates the setup values. In the case of the reef environment, there are two setup contributions, one where waves break over the reef and a second at the beach. When both slopes are included in the parameterisation, the fit improves further (not shown).

Finally, when analysing $R_{2\%}$ and comparing it to the complete parameterisation by Stockdon et al. (2006) for beaches, the fit improves considerably when the reef face slope is used instead of the beach face (Fig.6R). However, the runup parameterisations fail to predict the runup during extreme wave conditions. This is mainly ascribed to the underestimation of wave setup. Ongoing work is devoted to improving such parameterisations by incorporating the reef geometry characteristics in new runup parameterisations.

[Figure]

Fig. 3R a) Incident and b) infragravity swash parameterised in a dimensional form of the Iribarren equation and in comparison to Stockdon et al. (2006) (blue line) and a modified form, which includes the reef face slope (cyan line), and c) the parameterisation of Sig excluding the beach slope as suggested by Stockdon et al. (2006). Black dots represent all data, green the values associated with high water levels ($Z \geq Z15\% = 0.1636$ m) and red those associated with low water levels ($Z \leq Z15\% = -0.1636$ m).

[Figure]

Fig. 4R Ratio of incident to infragravity swash variance (v) against the Iribarren number. The solid line at log(v)=1 divides incident (above) from infragravity (below) dominated values. Black dots represent all data, green the values associated with high water levels ($Z \geq Z15\% = 0.1636$ m) and red those associated with low water levels ($Z \leq Z15\% = -0.1636$ m).

[Figure]

Fig. 5R a) wave setup parameterised in a dimensional form of the Iribarren equation and in comparison to Stockdon et al. (2006) (blue line) and a modified form, which includes the reef face slope (cyan line), and b) the parameterisation excluding the beach slope as suggested by Stockdon et al. (2006). Black dots represent all data, green the values associated with high water levels ($Z \geq Z15\%=0.1636$ m) and red those associated with low water levels ($Z \leq Z15\%=-0.1636$ m).

[Figure]

Fig. 6R Extreme runup values (R2%) for the 30 year hindcast data and the complete parameterisation suggested by Stockdon et al. (2006) with the beach face slope (blue line) and reef face slope (blue line). Black dots represent all data, green the values associated with high water levels ($Z \geq Z15\%=0.1636$ m) and red those associated with low water levels ($Z \leq Z15\%=-0.1636$ m).

**11. I like the inclusion of the dune height in the analysis but wonder if treating the dune as an un-erodible feature underestimates the overtopping.**

RESPONSE: The current model does not have the option for treating the dune nor the beach as erodible features, and is beyond the scope of the current study. However, a discussion on such limitations is now

included in the manuscript:

*"The present approach does not consider the dune or the beach as erodible features. Both play an important role in energy dissipation and hence further research is warranted to investigate their effects on wave overtopping."*

---

## Author Comment (AC2) · 8 Dec 2017

**Response to comments of Referee 2:**
**"The role of the reef-dune system in coastal protection in Puerto Morelos (Mexico)"**

(nhess-2017-304)

by Gemma L. Franklin, Alec Torres-Freyermuth, Gabriela Medellin, María Eugenia Allende-Arandia, Christian M. Appendini

**Referee #2:**

**The paper presents numerical modeling and analysis of wave runup on a reef-dune fronted coastline. The material presented is interesting and scientifically relevant. However, the description of methods and analysis of the data are lacking. Variables need to be defined more clearly and the notation (including italics) needs to be consistent throughout the text and figures; wave setup and 2% exceedance runup are two examples of inconsistent notation. The model validation with laboratory data is cursory and doesn't add much to the paper as it is written. The two model validation figures (Figures 2 and 3) do not state what runs are shown and whether it is a monochromatic or irregular wave case. This is extremely important as infragravity waves likely dominate the runup spectra in the field case. I would use the laboratory results and model comparison to highlight some of the important wave and wave setup dynamics, before going into the field modeling results. Of particular interest would be using the flume/ model results to explain the trends in Figure 8 and 10, which show less difference between reef degradation scenarios under large wave conditions. Not including storm surge is very problematic to the legitimacy of the hind-cast modeling. I think the authors should either include storm surge or reorganize the paper as more of a theoretical investigation rather than an applied hind-cast analysis. Major revisions are recommended before publication.**

RESPONSE: We thank the referee for his/her comments, which have helped improve the manuscript. The paper has been revised to be a more theoretical study. Furthermore, following the referees' comments we have: (i) conducted a more thorough analysis of runup dynamics and (ii) incorporated an analysis of the role of storm surge in the hindcast modelling as follows:

    (i)        Analysis of runup dynamics

Incident and infragravity swash height have been analysed using the parameterisations proposed by Stockdon et al., (2006). For beaches, these authors found incident swash height ($S_{inc}$) to be best parameterised by a dimensional version of an Iribarren–type relationship ($S_{inc}=0.75\beta(H_0L_0)^{1/2}$, where $\beta$ is the beach face slope, $H_0$ and $L_0$ incident wave height and length respectively. Fig. 1Ra shows the incident swash height for the present study (high and low water contributions are presented in green and red respectively). The 15% exceedance value of water level according to the astronomical tide $Z$ was used for high ($Z \geq Z15\%=0.1636$ m) and low water level ($Z \leq Z15\%=-0.1636$ m). As shown in the figure, Stockdon's parameterisation works fairly well for $S_{inc}$, particularly for high water levels, although it slightly overpredicts the numerical results. Figure 1Rb shows the results of using the same parameterisation for infragravity swash height ($S_{ig}$), as well as the effect of replacing the beach slope parameter ($\beta$) with the reef face slope ($\beta_{reef}$) (blue vs. cyan line), which results in an improved fit. Stockdon et al. (2006) found that by excluding beach slope in the parameterisation resulted in the best fit for $S_{ig}$ (Fig 1Rc), which also works fairly well for the high water level $S_{ig}$ values for the present study, although less applicable for more energetic waves. A notable difference between the runup contributions on reef-protected beaches with respect to sandy beaches is that $S_{ig}$ contributions were considerably larger. In order to look at this further, $S_{inc}$ vs. $S_{ig}$ variance was plotted against the Iribarren number (Fig. 2R), showing a clear dominance of $S_{ig}$ contributions under practically all wave conditions. This demonstrates a key difference in the swash contributions on beaches compared to reef environments, where

infragravity dominates.

With regards to wave setup <η>, the parameterisations presented by Stockdon et al. (2006), (a) with and (b) without beach face slope, underestimate wave setup for a reef environment (Fig. 3R). The effects of the relative contributions of high and low water to wave setup are less obvious for this profile than for sandy beaches (e.g. Medellin et al., 2016). When the slope of the reef face is used instead of the beach face slope, the parameterisation improves (cyan versus blue line Fig. 4Ra), although it still underestimates the setup values. In the case of the reef environment, there are two setup contributions, one where waves break over the reef and a second at the beach. When both slopes are included in the parameterisation, the fit improves further (not shown).

[Figure]

Fig. 1R a) Incident and b) infragravity swash parameterised in a dimensional form of the Iribarren equation and in comparison to Stockdon et al. (2006) (blue line) and a modified form, which includes the reef face slope (cyan line), and c) the parameterisation of $S_{ig}$ excluding the beach slope as suggested by Stockdon et al. (2006). Black dots represent all data, green the values associated with high water levels ($Z \geq Z15\% = 0.1636$ m) and red those associated with low water levels ($Z \leq Z15\% = -0.1636$ m).

[Figure]

Fig. 2R Ratio of incident to infragravity swash variance (v) against the Iribarren number. The solid line at log(v)=1 divides incident (above) from infragravity (below) dominated values. Black dots represent all data, green the values associated with high water levels (Z ≥ Z15%=0.1636 m) and red those associated with low water levels (Z ≤ Z15%=-0.1636 m).

[Figure]

Fig. 3R a) wave setup parameterised in a dimensional form of the Iribarren equation and in comparison to Stockdon et al. (2006) (blue line) and a modified form, which includes the reef face slope (cyan line), and b) the parameterisation excluding the beach slope as suggested by Stockdon et al. (2006). Black dots represent all data, green the values associated with high water levels (Z ≥ Z15%=0.1636 m) and red those associated with low water levels (Z ≤ Z15%=-0.1636 m).

[Figure]

Fig. 4R Extreme runup values for the 30 year hindcast data and the complete parameterisation suggested by Stockdon et al. (2006) with the beach face slope (blue line) and reef face slope (blue line). Black dots represent all data, green the values associated with high water levels (Z ≥ Z15%=0.1636 m) and red those associated with low water levels (Z ≤ Z15%=-0.1636 m).

(ii)      The role of storm surge in the hindcast modelling

The reason for not including the storm surge contribution is that the Hycom data only encompasses 16 years of the 30 years of data corresponding to the wave hindcast information. However, we also believe it is important to investigate its role using the available information. Therefore, the numerical model has been re-run selecting 300 representative cases, for the 16-year Hycom period (using the same methodology as for the 30 year hindcast), using both the sea surface height obtained from Hycom (including storm surge) and considering only the predicted tide. The numerical results made it possible to compare the effect of including this contribution on the storm impact scale. Figure 5R shows $R_{high}$ as a function of the return period while considering the two different scenarios. A significant increase in $R_{high}$ is observed when storm surge is included. This increase is important since it acts as a proxy for degradation, resulting in an underestimate of the effects of reef degradation on runup and hence coastal flooding when excluded. The effect of the storm surge (for the time period available) was smaller than the effect of the reef degrading by 1.1 m but slightly greater than the reef degrading by 0.3 m, particularly for return periods of less than 3 years. This will be incorporated in the discussion to highlight the fact that for the 30 year hindcast data, $R_{high}$ is underestimated by using the predicted tidal level, although this was all that was available in order to study a longer time period.

[Figure]

Fig, 5R Return value of $R_{high}$ for the model run with (open circles) and without (crosses) storm surge contribution for the time period of 1993-2008.

Furthermore, the paper was checked for consistency throughout the text and we removed the model validation section in accordance to the referees' suggestion.  A detailed point-by-point response to all the referee's comments are provided below.

**Specific comments**

**1. Figure 1: Label color bar.**

RESPONSE: This figure's colour bar has been labelled in the new version of the manuscript. Furthermore, the site location was modified to broaden the geographical setting (see figure 6R below).

[Figure]

Figure 6R. (a) Map of the study area in the Gulf of Mexico. The solid black line indicates the location of the bathymetric transect used in the numerical model. (b) Bathymetry obtained from the transect indicated on the map (bathymetry courtesy of CONABIO), including a beach profile surveyed in March, 2014 (courtesy of CINVESTAV-Merida).

**2. Figure 2: Need to state the wave forcing and still water level. Also, the measured wave setup is negative offshore due having a fixed volume of water in the flume (Figure 2b). The initial water levels in the model should be adjusted to this offshore water level and rerun with the correct offshore water level. I would include a low wave and a large wave example to highlight the dynamics seen in Figure 8 and 10.**

RESPONSE: In accordance with comments made by referees 1 and 2, the section on laboratory validation of the model, including Figures 2 and 3, has been removed in the revised version of the manuscript. Therefore, we have made reference to previous studies where the numerical model has been validated for reef profiles. The following text has been included in the revised manuscript:

*"The model is also capable of simulating wave-current interaction, wave breaking (Smit et al., 2013; de Bakker et al., 2015), and wave-runup (e.g., Brinkkemper et al., 2013; Ruju et al., 2014; Guimarao et al., 2015; Medellín et al., 2016). Furthermore, previous studies (Torres-Freyermuth et al., 2012; Zijlema et al., 2012; Buckley et al., 2014) have shown good agreement on simulating wave transformation on reef profiles."*

**3. Figure 4: Need to state the wave forcing and still water level. Are Z and Z_m the same? Include definition of Z and R_high in the caption.**

RESPONSE: We thank the referee for pointing out these issues. This was a typing mistake and has been corrected to Z in the text. The revised Figure's caption is as follows:

*"Figure 4. Reconstructed time series, including $R_{high}$, for the current reef profile using the 30-year hindcast wave conditions (wave height and period; $H_s$ and $T_p$) and predicted tidal level (Z). (a)–(c) Black lines indicate available hindcast data and red stars indicate the selected cases used to represent the complete time series. (d) Blue line represents time series reconstructed from the results of the simulated results. Red stars indicate the cases used for reconstruction. $R_{high} = R_{2\%} + Z$."*

**4. Figure 6: Is this data repeated in Figure 8? Remove this figure if it is.**

RESPONSE: We agree with the referee's comment and hence this figure has been removed from the manuscript since the data is also presented in Figure 8.

**5. Figure 8: This figure is extremely interesting. I would reorganize the paper to focus on explaining the trends seen here.**

RESPONSE: We thank the referee for bringing attention to this plot. One of the ecosystem services provided by the reef crest consists in the coastal protection via wave dissipation by breaking. The numerical results show that this becomes more important in the short to medium term storm events ($Tr$<10 years). On the other hand, above certain threshold conditions ($Tr$>10 years at this site) the reef-crest for the non-degraded condition no longer provided significant dissipation with respect to the degraded condition. The latter can be ascribed to the fact that the wave breaking point moves further offshore and that is why the degraded and non-degraded conditions present similar storm impact. This information is very important for insurance risk analysis. Therefore, following the referee's suggestion the structure, conclusions, and abstract will be re-organized to emphasize such trends.

**6. Page 2 Line 10: This paragraph needs revision. "The degradation of coral reefs affects the incident wave climate." Are you referring to the offshore wave climate? If so how does coral reef degradation affect offshore conditions?**

RESPONSE: This sentence refers to the conditions reaching the coast. Reef degradation affects wave transformation over the reef and hence the conditions that reach the shore. This sentence has been rewritten as:

*"The degradation of coral reefs affects the wave runup due to modifications in the spatial gradient of wave dissipation controlling both the incident wave energy and the wave-induced setup."*

**7. Page 3 Line 5-15: Most of this site description isn't needed.**

RESPONSE: We agree with the referee, hence most of this information has been removed in the revised manuscript. The following text was removed from the revised manuscript:

*"The climate in the region is hot and humid with a mean annual air temperature of 26.4°C, a maximum of 34.5°C in the summer and a minimum of 13°C in the winter (Merino and Otero, 1991). Rainfall is present all year round, although more intense during the summer, with a mean annual rainfall of 1,041 mm (Caribbean Coastal Marine Productivity Program: CARICOMP, unpublished data for the period 1993-1998). Evaporation varies from 102 mm in December to 178 mm in May (Merino and Otero 1983). The mean relative humidity is 84% (CONANP 2000). The water temperature at the bottom of the lagoon varies seasonally by around 5°C, from 31-32°C in August and September, to 24-25°C between December and March (Coronado et al., 2007)."*

**8. Page 4 Line 30: I would not include monochromatic waves in the model analysis and statistics as infragravity waves will be important for runup in the field.**

RESPONSE: The section on the laboratory experiment has been removed in accordance with suggestions made by Referees 1 and 2.

**9. Page 5 Line 5: I don't understand this discussion of bottom friction coefficients. Coefficients used in flume studies are discussed as if they are applicable to field cases? Are you using these values for your hind-cast analysis?**

RESPONSE: The friction coefficient was chosen based on reported values for numerical models used for coral reefs, for model validation. Friction in the field is likely to be much larger and, while important, the present study is not focused on this aspect. The focus of the present study is on the effects of vertical erosion of the reef and dune morphology more than on the effects of changes in roughness, which although important, require different scales to be considered and are beyond the scope of this study. Thus, we include the following sentences justifying the assumption:

*"This study focuses on the degradation of the reef-dune morphology and although reef roughness changes associated with degradation also play an important role in wave transformation (Franklin et al., 2013; Buckley et al., 2016) they are beyond the scope of the present work. In order to study its effects, high resolution Computational Fluid Dynamics (CFD) modelling is required to allow reef roughness to be taken into account explicitly (e.g., Osorio-Cano et al., sub judice)."*

**10. Page 5 Line 10: I would either remove the r^2 statistics or expand on how these values were calculated and which runs were used with table of runs, etc. I would probably remove and just focus on detailed analysis of two representative wave cases.**

RESPONSE: The section on the laboratory experiment has been removed in accordance with suggestions made by Referees 1 and 2.

**11. Page 5 Line 20: Did the flume have active reflection compensation? Were outgoing waves removed from the time series used to force the model? Generally need more details.**

RESPONSE: The wave flume is equipped with an active wave generation and absorption system developed by Aalborg University and VTI. However, the section on the laboratory experiment has been removed in accordance with suggestions made by Referees 1 and 2.

**12. Page 6 Line 25: In Figure 2 and elsewhere eta is used for wave setup. Here eta is a function of time. Change wave setup to overbar eta or <eta>. Sometimes 2% runup exceedance is written as R sometime Ru2% sometime Ru_2% sometime italics sometimes not. Make this consistent**

RESPONSE: We apologize for the inconsistencies in the variables definition. These terms have been corrected to ensure they are consistent throughout the paper. Ru2% is defined as $R_{2\%}$ and wave setup as $< \eta >$.

**13. Page 8 Line 20: I would go more into explaining the R_high response at larger wave heights. You mention resonance, but you don't provide evidence or state if this is the cause of the R_high response.**

RESPONSE: In the revised manuscript we include a more thorough discussion on the $R_{high}$ trends obtained for larger wave heights. The main reason for the difference in $R_{high}$ at larger wave heights is related to the role played by the reef in wave breaking. Under small wave heights, the reef plays an important role in this process, however as waves become larger, they break further offshore, and the reef no longer plays such an important role. When looking at the setup, swash and runup data, this change appears to take place for $H_oL_o^{1/2} > 30$ m (Fig.7R). Prior to this point there is a clear dominance in $S_{ig}$ and $R_{2\%}$ for the 1.1 m degraded scenario. For intermediate and large wave conditions, wave setup (Fig.7Ra) seems to be slightly greater for the conserved scenario as a result of the more intense wave breaking occurring over the reef crest and associated steeper gradient in radiation stress compared to the degraded scenario. However, for the degraded scenario the infragravity contribution is generally greater, where long waves enter the lagoon in the absence

of reflection at the reef (Fig.7Rc). The clear increase in $R_{2\%}$ for the degraded scenario demonstrated by Fig. 7Rd reiterates the importance of the reef in protecting the coast from flooding. This explains why at larger wave heights $R_{high}$ behaves as shown in Figure 8.

[Figure]

Figure 7R a) Wave setup, $\langle\eta\rangle$ b) incident swash ($S_{inc}$), c) infragravity swash ($S_{ig}$) and d) Extreme runup ($R_{2\%}$) against incident wave conditions. Black dots represent the setup data for the conserved reef profile, green the values the reef degraded 0.3 m and red those associated with the reef degraded 1.0 m. The vertical arrow indicates the change in behaviour.

**14. Page 10 Line 1: The Buckley et al. 2015 reference should be Buckley et al. 2016**

RESPONSE: This reference has been corrected. The reference was changed to Buckley et al. 2016 and was included in the reference list.

---

## Author Comment (AC3) · 8 Dec 2017

**Response to comments of Referee 3: "The role of the reef-dune system in coastal protection in Puerto Morelos (Mexico)"**

(nhess-2017-304)

by Gemma L. Franklin, Alec Torres-Freyermuth, Gabriela Medellin, María Eugenia Allende-Arandia, Christian M. Appendini

**Referee #3:**

The manuscript presents a numerical study on the role that a reef-dune system plays in protecting a given coast from storms. The case study of Puerto Morelos, Mexico has been selected to this end. The study illustrates the importance of a holistic management of the coast (considering the reef and dunes as part of a single system) in order to maximise the protective service obtained from ecosystems, which is very relevant in the context of coastal developments, climate change and other factors that compromise the stability of such habitats. Therefore, the paper may potentially be very useful to policy makers, engineers and scientists concerned with a sustainable management of the coast. However, the study also presents some significant weaknesses that should be amended before publication of the manuscript is advised. Please find below a list of points –in decreasing order of importance- that should be addressed by the authors before I can recommend publication of the present paper in NHESS.

**RESPONSE**: We thank the referee for his/her comments, which have helped us improve the manuscript. A detailed point-by-point response to the referee's concerns is provided below. Furthermore, following the main referees' comments we have: (i) conducted a more thorough analysis of runup dynamics and (ii) incorporated an analysis of the role of storm surge in the hindcast modelling as follows:

(i) Analysis of runup dynamics

Incident and infragravity swash height have been analysed using the parameterisations proposed by Stockdon et al., (2006). For beaches, these authors found incident swash height ( $S_{inc}$ ) to be best parameterised by a dimensional version of an Iribarren-type relationship ( $S_{inc}=0.75\beta(H_0L_0)^{1/2}$ , where  $\beta$  is the beach face slope,  $H_0$ and  $L_0$  incident wave height and length respectively. Fig. 1Ra shows the incident swash height for the present study (high and low water contributions are presented in green and red respectively). The 15% exceedance value of water level according to the astronomical tide Z was used for high ( $Z \ge Z15\%=0.1636$  m) and low water level ( $Z \le Z15\%$ =-0.1636 m). As shown in the figure, Stockdon's parameterisation works fairly well for Sinc, particularly for high water levels, although it slightly over predicts the numerical results. Figure 1Rb shows the results of using the same parameterisation for infragravity swash height  $(S_{ig})$ , as well as the effect of replacing the beach slope parameter ( $\beta$ ) with the reef face slope ( $\beta_{reef}$ ) (blue vs. cyan line), which results in an improved fit. Stockdon et al. (2006) found that by excluding beach slope in the parameterisation resulted in the best fit for Sig (Fig 1Rc), which also works fairly well for the high water level Sig values for the present study, although less applicable for more energetic waves. A notable difference between the runup contributions on reef-protected beaches with respect to sandy beaches is that Sig contributions were considerably larger. In order to look at this further,  $S_{\text{inc}}$  vs.  $S_{\text{ig}}$  variance was plotted against the Iribarren number (Fig. 2R), showing a clear dominance of Sig contributions under practically all wave conditions. This demonstrates a key difference in the swash contributions on beaches compared to reef environments, where infragravity dominates.

With regards to wave setup  $\langle \eta \rangle$ , the parameterisations presented by Stockdon et al. (2006), with (a) and without (b) beach face slope, underestimate wave setup for a reef environment (Fig. 3R). The effects of the relative contributions of high and low water to wave setup are less obvious for this profile than for sandy beaches (e.g. Medellin et al., 2016). When the slope of the reef face is used instead of the beach face slope,

the parameterisation improves (cyan versus blue line Fig. 3Ra), although it still underestimates the setup values. In the case of the reef environment, there are two setup contributions, one where waves break over the reef and a second at the beach. When both slopes are included in the parameterisation, the fit improves further (not shown).

Finally, when analysing  $R_{2\%}$  and comparing it to the complete parameterisation by Stockdon et al. (2006) for beaches, the fit improves considerably when the reef face slope is used instead of the beach face (Fig. 4R). However, the runup parameterisations fail to predict the runup during extreme wave conditions. This is mainly ascribed to the underestimation of wave setup. Ongoing work is devoted to improving such parameterizations by incorporating the reef geometry characteristics.

Fig. 1R a) Incident and b) infragravity swash parameterised in a dimensional form of the Iribarren equation and in comparison, to Stockdon et al. (2006) (blue line) and a modified form, which includes the reef face slope (cyan line), and c) the parameterisation of  $S_{ig}$  excluding the beach slope as suggested by Stockdon et al. (2006). Black dots represent all data, green the values associated with high water levels ( $Z \ge Z15\%$ =0.1636 m) and red those associated with low water levels ( $Z \le Z15\%$ =-0.1636 m).

Fig. 2R Ratio of incident to infragravity swash variance (v) against the Iribarren number. The solid line at log(v)=1 divides incident (above) from infragravity (below) dominated values. Black dots represent all data, green the values associated with high water levels ( $Z \ge Z15\%=0.1636$  m) and red those associated with low water levels ( $Z \le Z15\%=0.1636$  m).

---

## Author Response (AR1)

**Response to comments of Referee 1:**
**"The role of the reef-dune system in coastal protection in Puerto Morelos (Mexico)"**

(nhess-2017-304)

by Gemma L. Franklin, Alec Torres-Freyermuth, Gabriela Medellin, María Eugenia Allende-Arandia, Christian M. Appendini

**Referee #1:**
**Review of "The role of reef-dune systems in coastal protection in Puerto Morelos (Mexico)" by Franklin et al. This paper presents an analysis of the combined impact of reef and dune degradation on determining storm impact. In general I found the paper interesting and conclusions primarily supported (and very timely given recent events), but the analysis a bit lacking. See my detailed comments below, but my general recommendation is that this paper needs a major revision prior to publication. The numerical simulations conducted by the authors can provide much more information about what is causing the observed runup extremes and it would be good to delve a bit deeper into what is going on.**

RESPONSE: We thank the referee for his/her comments which have helped us improve the manuscript. A detailed point-by-point response to the referee's comments is provided below. Following the referee's suggestion the revised manuscript includes: (i) a more thorough analysis on the role that storm surge has in the storm impact, and (ii) a detailed study of runup dynamics.

**Specific comments**

**1. Pg 3, lines 5-15. A majority of this information is not relevant, e.g. annual temperature and rainfall do not impact the runup.**

RESPONSE: We agree with the referee and hence most of this information has been removed from this section in the revised manuscript. The following text has been removed from the manuscript:

*"The climate in the region is hot and humid with a mean annual air temperature of 26.4°C, a maximum of 34.5°C in the summer and a minimum of 13°C in the winter (Merino and Otero, 1991). Rainfall is present all year round, although more intense during the summer, with a mean annual rainfall of 1,041 mm (Caribbean Coastal Marine Productivity Program: CARICOMP, unpublished data for the period 1993-1998). Evaporation varies from 102 mm in December to 178 mm in May (Merino and Otero 1983). The mean relative humidity is 84% (CONANP 2000). The water temperature at the bottom of the lagoon varies seasonally by around 5°C, from 31-32°C in August and September, to 24-25°C between December and March (Coronado et al., 2007)."*

*"The lagoon bed is characterised by calcareous sand covered by patches of seagrass with occasional coral colonies."*

*"The mean grain size of the sediment is 0.3 mm, and the mean lower beach slope is 0.05 (Ruiz de Alegría-Arazaburu et al., 2013)."*

**2. Pg. 4, line 10. Please specify which model.**

RESPONSE: The name of the model and reference have been included (page 4, lines 19-20 in the revised manuscript). The text has been changed to:

*"These data were estimated using the third-generation spectral wave model MIKE 21 SW (Sørensen et al. 2004) forced with wind data from the North American Regional Reanalysis (NARR) (Mesinger et al., 2006)."*

**3. Pg. 4, line 13. A high r2 does not indicate model performance unless coupled with the regression slope. The r2 only tells you how well a model reproduces the variance.**

RESPONSE: Further information on the model performance has been included (page 4, lines 22-24 in the revised manuscript).

*"The mean observed wave height ($H_s$) and peak period ($T_p$) were 1.22 m and 6.70 s respectively, compared to the mean reanalysis/hindcast values of 1.31 m and 7.27 s (Rms of 0.33 $H_s$ and 1.59 for $T_p$ with correlation coefficients of 0.90 and 0.51)."*

**4. Section 3. I do not see the point in including the flume experiments in this paper. You are essentially calibrating the model on an unrelated data set for a reef/beach profile that was not made to replicate your field site. Essentially you are just showing that SWASH works on reef profiles which has already been shown (Zijlema, 2012, Buckley et al., 2014). Additionally, and while it is unfortunately the case, showing that the model is calibrated at one site does not mean it is calibrated at all other sites. As a result, my preference would be to entirely remove the discussion and comparison with the flume results and use the extra space to further develop the results as they result to the field site. Also I found the discussion of the runs with and without the reef crest confusing.**

RESPONSE: The referee's comment is consistent with those from the other two referees in the sense that there is no added value by including the flume experiments in this paper. Therefore, the section on the model validation employing laboratory data has been removed. One of the co-authors has been removed from the list of authors since his contribution was related to the laboratory experiments. In the revised manuscript we have made reference to previous studies where the model has been validated for reef profiles. The following text has been included in the revised manuscript (page 4, line 31-page 5, line 4 in the revised manuscript):

*"The model is also capable of simulating wave-current interaction, wave breaking (Smit et al., 2013; de Bakker et al., 2015), wave transformation on reefs (e.g. Torres-Freyermuth et al., 2012; Zijlema et al., 2012; Buckley et al., 2014), and wave-runup (e.g., Brinkkemper et al., 2013; Ruju et al., 2014; Guimarao et al., 2015; Medellín et al., 2016). Therefore this numerical model is suitable for conducting a numerical study on wave transformation and wave runup in the Puerto Morelos reef lagoon. For further model details, including model equations see Zijlema et al. (2011)."*

**5. Figure 1. Can the inset be made a higher resolution and zoomed out a bit to provide more geographic context?**

RESPONSE: Following the referee's suggestions Figure 1 has been modified to provide a better geographical context and improve its quality.

**6. How long is each simulation run for? This is important in determining the validity of the statistics which include long waves.**

RESPONSE: The SWASH simulations were run for 3200 s which were sampled for 2700 s (see page 7, line 31) after accounting for model spin-up time (500 s), considering a constant water level. The simulation time is sufficient to account for long wave statistics.

**7. Page 6 line 26. Extreme runup is defined inconsistently.**

RESPONSE: We thank the referee for pointing out this inconsistency. Extreme value statistics of runup, often defined as the elevation exceeded by only 2% of runup, is denoted by $R_{2\%}$ (Holman, 1986).

**8. Page 6. Line 30. I am confused about the definition of Rlow. The setup is the average runup so why is this no Ravg? Also it would be helpful to remind the reader that here Z is the tidal level.**

RESPONSE: Here, $R_{low}$ is adopted from a previous study (i.e., Sallenger, 2000) where the storm impact scale was introduced for the first time. The term $R_{low}$ consists of the average runup (setup) plus the tidal level and, where applicable, the storm surge. Thus, it represents the low extreme sea level resulting from the aforementioned contributions, which is then compared with morphological features in the beach profile. Therefore, we have maintained this terminology for consistency with previous work (e.g., Stockdon et al., 2007; Medellín et al., 2016; among many others). The revised manuscript includes the following text for further clarification (page 8, lines 8-9):

*"$R_{low}$ represents the low extreme sea level resulting from the setup, tidal level and storm surge contributions consistent with Sallenger (2000)."*

**9. Page 6 line 32. As is sort of acknowledged in the discussion, not including surge is a huge limitation of the approach. As the depth of reef submergence directly effects the short wave transmission across the reef the surge is critical in determining the runup (in addition to the fact that the surge adds to the water level from which waves runup). Could you not include this for the simulations suing the hycom model? I find this a major limitation of the current study. High surge also acts as a proxy for the reef degradation, and thus neglecting surge probably causes your results to underestimate the occurrence of over toping.**

RESPONSE: We agree with the referee on the importance of including the storm surge. The reason for not including the storm surge contribution in the previous version of the manuscript is the lack of long tidal records in the study area and that the Hycom data only encompasses 16 years of the 30 years of data corresponding to the wave hindcast information. However, we recognise the importance of studying the role of the storm surge using the available information. Therefore, the numerical model has been re-run selecting 300 representative cases, for the 16-year Hycom period (using the same methodology as for the 30-year hindcast), using: (i) the sea surface height obtained from Hycom (mean sea level including storm surge and astronomical tide) and (ii) the astronomical tide (see Section 5.4 in the revised manuscript - page 11, line 23 to page 12, line 18). The numerical results made it possible compare the effect of including this contribution on the extreme water levels and the storm impact.

The following text has been included in the revised manuscript (page 11, line 23-page 12, line 18):

*5.4 Role of Storm surge*

*To investigate the storm surge contribution, sea level data were obtained from the HYbrid Coordinate Ocean Model (HYCOM; Halliwell et al., 1998; Bleck, 2001) for the Gulf of Mexico (GoM) (https://hycom.org/data/goml0pt04) for the dates that coincided with the available wave hindcast information (1993-2008). For the GoM, HYCOM has a 1/25° or 0.04° equatorial and latitudinal resolution (~3.5 km) for each variable at mid-latitudes. The version of HYCOM used is 2.2.77. Both $H_s$ and $T_p$ from the Hindcast data were interpolated to the same time vector as that of the GoM sea level data. A total of 300 representative cases were simulated for the 16-year period (using the same methodology as for the 30-year hindcast), using: (i) the sea surface height obtained from Hycom (mean sea level including storm surge and astronomical tide) or (ii) the astronomical tide. Figure 7 shows $R_{high}$ as a function of the return period while considering the two different scenarios. An increase in $R_{high}$ is observed when storm surge is included. This increase is important since it acts as a proxy for reef degradation. Neglecting the storm surge contribution results in an underestimate of the effects of reef degradation on runup and hence coastal flooding. However, the effect of the storm surge (for the time period available) was smaller than the effect of the reef degrading by 1.1 m but slightly greater than the reef degrading by 0.3 m, particularly for return periods of less than 3 years (Fig. 7).*

*In order to study the effects of the storm surge on extreme water levels for the specific case of a hurricane event, wave parameters were selected from the hindcast data between the 19[th] and the 25[th] of October (Fig. 8a and b), corresponding to Hurricane Wilma, a Category 5 hurricane, which reached the*

*Yucatan Peninsula on the 20th-21st of October, 2005. The maximum values are higher and the minimum values are lower owing to the storm surge contribution during the hurricane passage. In terms of reef degradation and the effects of the storm surge during the hurricane, the $R_{high}$ values are generally greater for the degraded profiles throughout the five days presented, except around the peak of the hurricane (results not shown). This might be ascribed to waves breaking further offshore of the reef crest. Therefore, the storm impact during more extreme conditions appears to be less sensitive to reef crest degradation than during moderate storm conditions, further supporting the reef degradation results presented in Section 5.2. It is also important to note that during an extreme event, such as Hurricane Wilma, the reef can act as a barrier against sediment transport, further reducing the storm impact on the coast by retaining sand in the lagoon and on the beach. However, this is not taken into account in the present study, nor is the effect of changes in reef roughness associated with degradation, which have been shown to have important implications in wave transformation (Buckley et al., 2016) and wave runup (Osorio et al., 2017) but are not the focus of the present study. Furthermore, it is likely that by treating the dune as a non-erodible feature, overtopping is underestimated.*

**10. I think the results section could be considerably beefed up. By using a phase resolving model you allow for a lot of information on the runup dynamics to be gleaned. As has been demonstrated in the available literature reef/lagoon systems can often act as open basins and thus have the potential to enhance/trap IG energy.**

RESPONSE: By removing the section on the laboratory validation, there is more room to look into the results in greater detail. Thus, runup was separated into the incident ($S_{inc} = fp*0.5 < S \leq fp*2$) and infragravity ($S_{ig} = fp*0.1 < S \leq fp*0.5$) swash frequencies (Figs. 6b and c) and setup (Fig. 6a). Furthermore, setup, swash and runup data were analysed in further detail (Section 5.2, page 10, line 24-33):

*In order to explain the observed differences in Rhigh at larger wave heights, the runup was separated into the incident ($S_{inc} = fp*0.5 < S < fp*2$) and infragravity ($S_{ig} = fp*0.1 < S < fp*0.5$) swash frequencies (Figs. 6b and c) and setup (Fig. 6a). Furthermore, setup, swash and runup data were analysed in further detail. The change in the importance of the reef crest in the wave breaking process seems to take place for $H_0 L_0{}^{1/2} > 30$ m (Fig. 6). Prior to this point there is a clear dominance in $S_{ig}$ and $R_{2\%}$ for the 1.1 m degraded scenario. For intermediate and large wave conditions, wave setup (Fig. 6a) seems to be slightly greater for the non-degraded scenario as a result of the more intense wave breaking occurring over the reef crest and associated steeper gradient in radiation stress compared to the degraded scenario. However, for the degraded scenario the infragravity contribution is generally greater, where long waves enter the lagoon in the absence of reflection at the reef (Fig.6c). The clear increase in $R_{2\%}$ for the degraded scenario demonstrated by Figure 6d reiterates the importance of the reef in protecting the coast from flooding.*

In addition a discussion on the applicability of current runup parameterisations to a reef environment is presented in Section 6 (page 12, line 19 – page 14, line 14):

*The calculation of extreme runup is necessary to estimate the storm impact in coastal areas. Runup parameterisations provide a rapid assessment of coastal vulnerability and hence deserve further investigation. Under certain combinations of energetic wave conditions on fringing reefs, the steep reef face has been shown to facilitate the liberation of fluctuations with infragravity periods, which can pass into the lagoon with little energy loss and exacerbate the effect of the storm (Roeber et al., 2015). The importance of these long-wave motions inside the lagoon has been previously demonstrated by Van Dongeren et al. (2013). The above phenomenon can be intensified if the reef lagoon resonates with the wave period, amplifying the peak energy of the surf beat (Torres-Freyermuth et al., 2012; Roeber et al., 2015). Therefore, runup dynamics and the validity of applying parameterisations used for beaches in reef environments are investigated here.*
*        Incident and infragravity swash height have been analysed for the conserved scenario using the parameterisations proposed by Stockdon et al., (2006) where the swash height was calculated as follows:*

$$S = \sqrt{(S_{inc})^2 + (S_{ig})^2}$$

*where $S_{inc}$ and $S_{ig}$ are significant swash height in the incident and infragravity frequencies respectively. For*

*beaches, Stockdon et al. (2006) found incident swash height ($S_{inc}$) to be best parameterised by a dimensional version of an Iribarren–type relationship, $S_{inc}=0.75\beta(H_0L_0)^{1/2}$, where β is the beach face slope, $H_0$ and $L_0$ incident wave height and length respectively. Figure 9a shows the incident swash height for the 600 cases simulated in the present study (high and low water contributions are presented in green and red respectively). As shown in the figure, Stockdon's parameterisation (blue solid line) works fairly well for $S_{inc}$, particularly for high water levels, although it slightly overpredicts the numerical results. Figure 8b shows the parameterisation for infragravity swash height ($S_{ig}$), excluding beach slope in the parameterisation, which also works satisfactorily for the high-water level, although is less applicable for more energetic waves. A notable difference between the runup contributions on reef-protected beaches with respect to sandy beaches is that $S_{ig}$ contributions were considerably larger. In order to look at this further, $S_{inc}$ vs. $S_{ig}$ variance was plotted against the Iribarren number (not shown), showing a clear dominance of $S_{ig}$ contributions under practically all wave conditions. This demonstrates a key difference in the swash contributions on beaches compared to reef environments, where infragravity dominates irrespective of the beach slope conditions.*

*With regards to wave setup <η>, the parameterisations presented by Stockdon et al. (2006) significantly underestimate wave setup in the study area (Fig. 9c). The effects of the relative contributions of high and low water to wave setup are less obvious for this profile than for sandy beaches (e.g. Medellin et al., 2016). When the slope of the reef face is used instead of the beach face slope, the parameterisation improves (red versus blue line Fig. 9c), although it still underestimates the setup values.*

*Finally, when analysing $R_{2\%}$ and comparing it to the complete parameterisation by Stockdon et al. (2006) for beaches, the fit improves considerably when the reef face slope is used instead of the beach face (Fig. 10). However, the runup parameterisations fail to predict the runup during extreme wave conditions. This is mainly attributed to the underestimation of wave setup. However it is worth noticing that the good fit of the $R_{2\%}$ parameterisation is ascribed to a combination of the over prediction of S and under prediction of setup. Therefore, future work should be devoted to improving such parameterisations by incorporating the reef geometry characteristics in the formulations.*

**11. I like the inclusion of the dune height in the analysis but wonder if treating the dune as an unerodible feature underestimates the overtopping.**

RESPONSE: The current model does not have the option for treating the dune nor the beach as erodible features, and this approach is beyond the scope of the current study. However, a discussion on such limitations is now included in the manuscript (page 14 line 32-page 15, line 4):

*However, the main drawback in the present study is that it does not consider the dune or the beach as erodible features. Both play an important role in energy dissipation and hence further research is warranted to investigate its effects on increasing/decreasing the storm impact during extreme events. Furthermore, the role of reef roughness and two-dimensional horizontal processes need to be addressed for a more comprehensive study on the implication of reef degradation in such environments*

**Response to comments of Referee 2:**
**"The role of the reef-dune system in coastal protection in Puerto Morelos (Mexico)"**

(nhess-2017-304)

by Gemma L. Franklin, Alec Torres-Freyermuth, Gabriela Medellin, María Eugenia Allende-Arandia, Christian M. Appendini

**Referee #2:**

**The paper presents numerical modeling and analysis of wave runup on a reef-dune fronted coastline. The material presented is interesting and scientifically relevant. However, the description of methods and analysis of the data are lacking. Variables need to be defined more clearly and the notation (including italics) needs to be consistent throughout the text and figures; wave setup and 2% exceedance runup are two examples of inconsistent notation. The model validation with laboratory data is cursory and doesn't add much to the paper as it is written. The two model validation figures (Figures 2 and 3) do not state what runs are shown and whether it is a monochromatic or irregular wave case. This is extremely important as infragravity waves likely dominate the runup spectra in the field case. I would use the laboratory results and model comparison to highlight some of the important wave and wave setup dynamics, before going into the field modeling results. Of particular interest would be using the flume/ model results to explain the trends in Figure 8 and 10, which show less difference between reef degradation scenarios under large wave conditions. Not including storm surge is very problematic to the legitimacy of the hind-cast modeling. I think the authors should either include storm surge or reorganize the paper as more of a theoretical investigation rather than an applied hind-cast analysis. Major revisions are recommended before publication.**

RESPONSE: We thank the referee for his/her comments, which have helped improve the manuscript. The paper has been revised to be a more theoretical study. We have also included additional information in the methods and analysis (see Section 4.1, page 6, line 29 - page 7, line 12 and Section 4.2 page 8, line 15-20). Much of this information was omitted in the previous version of the manuscript since it is available in the cited literature.

Furthermore, following the referees' comments the revised manuscript contains (i) a more thorough analysis on the role that storm surge has in the storm impact, and (ii) a detailed study of runup dynamics. The reason for not including the storm surge contribution in the previous version of the manuscript is the lack of long tidal records in the study area and that the Hycom data only encompasses 16 years of the 30 years of data corresponding to the wave hindcast information. However, we recognise the importance of studying the role of the storm surge using the available information. Therefore, the numerical model has been re-run selecting 300 representative cases, for the 16-year Hycom period (using the same methodology as for the 30-year hindcast), using: (i) the sea surface height obtained from Hycom (mean sea level including storm surge and astronomical tide) and (ii) the astronomical tide (see Section 5.4 in the revised manuscript - page 11, line 23 to page 12, line 18). The numerical results made it possible compare the effect of including this contribution on the extreme water levels and the storm impact.

The following text has been included in the revised manuscript (page 11, line 23-page 12, line 18):

[revised manuscript text omitted]

Furthermore, the paper was checked for consistency throughout the text and we have removed the model validation section in accordance to the referees' suggestion. A detailed point-by-point response to all the referee's comments are provided below.

**Specific comments**

**1. Figure 1: Label color bar.**

RESPONSE: This figure's colour bar has been labelled in the new version of the manuscript. Furthermore, the site location was modified to broaden the geographical setting.

**2. Figure 2: Need to state the wave forcing and still water level. Also, the measured wave setup is negative offshore due having a fixed volume of water in the flume (Figure 2b). The initial water levels in the model should be adjusted to this offshore water level and rerun with the correct offshore water level. I would include a low wave and a large wave example to highlight the dynamics seen in Figure 8 and 10.**

RESPONSE: In accordance with comments made by referees 1 and 2, the section on laboratory validation of the model, including Figures 2 and 3, has been removed in the revised version of the manuscript. Therefore, the author responsible for these contributions has also been removed from the list of authors. In the revised manuscript we have made reference to previous studies where the model has been validated for reef profiles. The following text has been included in the revised manuscript (page 4, line 31-page 5, line 4 in the revised manuscript):

*"The model is also capable of simulating wave-current interaction, wave breaking (Smit et al., 2013; de Bakker et al., 2015), wave transformation on reefs (e.g. Torres-Freyermuth et al., 2012; Zijlema et al., 2012; Buckley et al., 2014), and wave-runup (e.g., Brinkkemper et al., 2013; Ruju et al., 2014; Guimarao et al., 2015; Medellín et al., 2016). Therefore this numerical model is suitable for conducting a numerical study on wave transformation and wave runup in the Puerto Morelos reef lagoon. For further model details, including model equations see Zijlema et al. (2011)."*

**3. Figure 4: Need to state the wave forcing and still water level. Are Z and Z_m the same? Include definition of Z and R_high in the caption.**

RESPONSE: We thank the referee for pointing out these issues. This was a typing mistake and has been corrected to Z in the text. The revised Figure's caption (now Figure 2) is as follows:

*"Figure 2. Reconstructed time series, including the extreme water level $R_{high}$, for the current reef profile using the 30-year hindcast wave conditions (wave height and period; $H_s$ and $T_p$) and astronomical tide (Z). (a)–(c) Black lines indicate available hindcast data and red stars indicate the selected cases used to represent the complete time series. (d) Blue line represents time series reconstructed from the results of the simulated results. Red stars indicate the cases used for reconstruction. $R_{high}=R_{2\%}+Z$."*

**4. Figure 6: Is this data repeated in Figure 8? Remove this figure if it is.**

RESPONSE: We agree with the referee's comment and hence this figure has been removed from the manuscript since the data was also presented in Figure 8 (now Figure 5 in the revised manuscript).

**5. Figure 8: This figure is extremely interesting. I would reorganize the paper to focus on explaining the trends seen here.**

RESPONSE: We thank the referee for bringing attention to this plot (now Figure 5 in the revised manuscript). One of the ecosystem services provided by the reef crest consists in the coastal protection via wave dissipation by breaking. The numerical results show that this becomes more important in the short to medium term storm events ($Tr$<10 years). On the other hand, above certain threshold conditions ($Tr$>10 years at this site) the reef-crest for the non-degraded condition no longer provided significant dissipation with respect to the degraded condition. The latter can be ascribed to the fact that the wave breaking point moves further offshore and that is why the degraded and non-degraded conditions present similar storm impact. This information is very important for insurance risk analysis. Therefore, following the referee's suggestion the structure, conclusions, and abstract have been re-organized to emphasize such trends.

**6. Page 2 Line 10: This paragraph needs revision. "The degradation of coral reefs affects the incident wave climate." Are you referring to the offshore wave climate? If so how does coral reef degradation affect offshore conditions?**

RESPONSE: This sentence refers to the conditions reaching the coast. Reef degradation affects wave transformation over the reef and hence the conditions that reach the shore. This sentence has been rewritten (page 2, line 17-18) as:

*"The degradation of coral reefs affects the wave runup due to modifications in the spatial gradient of wave dissipation, controlling both the incident swash and wave-induced setup."*

**7. Page 3 Line 5-15: Most of this site description isn't needed.**

RESPONSE: We agree with the referee, hence most of this information has been removed in the revised manuscript. The following text was removed from the revised manuscript:

*"The climate in the region is hot and humid with a mean annual air temperature of 26.4°C, a maximum of 34.5°C in the summer and a minimum of 13°C in the winter (Merino and Otero, 1991). Rainfall is present all year round, although more intense during the summer, with a mean annual rainfall of 1,041 mm (Caribbean Coastal Marine Productivity Program: CARICOMP, unpublished data for the period 1993-1998). Evaporation varies from 102 mm in December to 178 mm in May (Merino and Otero 1983). The mean relative humidity is 84% (CONANP 2000). The water temperature at the bottom of the lagoon varies seasonally by around 5°C, from 31-32°C in August and September, to 24-25°C between December and March (Coronado et al., 2007)."*

*"The lagoon bed is characterised by calcareous sand covered by patches of seagrass with occasional coral colonies."*

*"The mean grain size of the sediment is 0.3 mm, and the mean lower beach slope is 0.05 (Ruiz de Alegría-Arazaburu et al., 2013)."*

**8. Page 4 Line 30: I would not include monochromatic waves in the model analysis and statistics as infragravity waves will be important for runup in the field.**

RESPONSE: The section on the laboratory experiment has been removed in accordance with suggestions made by Referees 1 and 2.

**9. Page 5 Line 5: I don't understand this discussion of bottom friction coefficients. Coefficients used in flume studies are discussed as if they are applicable to field cases? Are you using these values for your hind-cast analysis?**

RESPONSE: The friction coefficient was chosen based on reported values for numerical models used for coral reefs, for model validation. Friction in the field is likely to be much larger and, while important, the present study is not focused on this aspect. The focus of the present study is on the effects of vertical erosion of the reef and dune morphology more than on the effects of changes in roughness, which although important, require different scales to be considered and are beyond the scope of this study. Thus, we include the following sentences (page 5, line 20-25) justifying the assumption:

*"Although likely to be lower than values obtained in field studies, in the absence of measured values for the study site, this coefficient was used in the numerical simulations. This study focuses on the degradation of the reef-dune morphology. Reef roughness changes also play an important role in wave transformation (Franklin et al., 2013; Buckley et al., 2016) but high resolution Computational Fluid Dynamics (CFD) modelling is required to allow reef roughness to be taken into account explicitly (e.g. Osorio-Cano et al., sub judice). Therefore the study of these effects is beyond the scope of the present work"*

**10. Page 5 Line 10: I would either remove the r^2 statistics or expand on how these values were calculated and which runs were used with table of runs, etc. I would probably remove and just focus on detailed analysis of two representative wave cases.**

RESPONSE: The section on the laboratory experiment has been removed in accordance with suggestions made by Referees 1 and 2.

**11. Page 5 Line 20: Did the flume have active reflection compensation? Were outgoing waves removed from the time series used to force the model? Generally need more details.**

RESPONSE: The wave flume is equipped with an active wave generation and absorption system developed by Aalborg University and VTI. However, the whole section on the laboratory experiment has been removed in accordance with suggestions made by Referees 1 and 2.

**12. Page 6 Line 25: In Figure 2 and elsewhere eta is used for wave setup. Here eta is a function of time. Change wave setup to overbar eta or <eta>. Sometimes 2% runup exceedance is written as R sometime Ru2% sometime Ru_2% sometime italics sometimes not. Make this consistent.**

RESPONSE: We apologize for the inconsistencies in the variables definition. These terms have been corrected to ensure they are consistent throughout the manuscript. Ru2% is defined as $R_{2\%}$ and wave setup as $<\eta>$.

**13. Page 8 Line 20: I would go more into explaining the R_high response at larger wave heights. You mention resonance, but you don't provide evidence or state if this is the cause of the R_high response.**

RESPONSE: In the revised manuscript we include a more thorough discussion on the $R_{high}$ trends obtained for larger wave heights. The main reason for the difference in $R_{high}$ at larger wave heights is related to the role

played by the reef in wave breaking. The following text and Figure 6 have been added to better explain these trends (page 10, line 18- line 33):

*The behaviour of $R_{high}$ for larger wave heights is related to the role played by the reef in wave breaking. Under small wave heights, the reef plays an important role in this process, however, as waves become larger they break further offshore than the location of the reef crest, hence the reef no longer plays such an important role. This seems to occur for return periods of approximately 10 years or greater. Furthermore, the larger the waves, the more the water depth will increase due to wave setup, making the differences in $R_{high}$ due to reef degradation less noticeable*

*In order to explain the observed differences in $R_{high}$ at larger wave heights, the runup was separated into the incident ($S_{inc} = fp*0.5<S<fp*2$) and infragravity ($S_{ig} = fp*0.1<S<fp*0.5$) swash frequencies (Figs. 6b and c) and setup (Fig. 6a). Furthermore, setup, swash and runup data were analysed in further detail. The change in the importance of the reef crest in the wave breaking process seems to take place for $H_0L_0^{^1/2} >30$ m (Fig. 6). Prior to this point there is a clear dominance in $S_{ig}$ and $R_{2\%}$ for the 1.1 m degraded scenario. For intermediate and large wave conditions, wave setup (Fig. 6a) seems to be slightly greater for the non-degraded scenario as a result of the more intense wave breaking occurring over the reef crest and associated steeper gradient in radiation stress compared to the degraded scenario. However, for the degraded scenario the infragravity contribution is generally greater, where long waves enter the lagoon in the absence of reflection at the reef (Fig. 6c). The clear increase in $R_{2\%}$ for the degraded scenario demonstrated by Figure 6d reiterates the importance of the reef in protecting the coast from flooding.*

**14. Page 10 Line 1: The Buckley et al. 2015 reference should be Buckley et al. 2016**

RESPONSE: This reference has been corrected. The reference was changed to Buckley et al. 2016 and was included in the reference list.

**Response to comments of Referee 3:**
**"The role of the reef-dune system in coastal protection in Puerto Morelos (Mexico)"**

(nhess-2017-304)

by Gemma L. Franklin, Alec Torres-Freyermuth, Gabriela Medellin, María Eugenia Allende-Arandia, Christian M. Appendini

**Referee #3:**

**The manuscript presents a numerical study on the role that a reef-dune system plays in protecting a given coast from storms. The case study of Puerto Morelos, Mexico has been selected to this end. The study illustrates the importance of a holistic management of the coast (considering the reef and dunes as part of a single system) in order to maximise the protective service obtained from ecosystems, which is very relevant in the context of coastal developments, climate change and other factors that compromise the stability of such habitats. Therefore, the paper may potentially be very useful to policy makers, engineers and scientists concerned with a sustainable management of the coast. However, the study also presents some significant weaknesses that should be amended before publication of the manuscript is advised. Please find below a list of points –in decreasing order of importance- that should be addressed by the authors before I can recommend publication of the present paper in NHESS.**

RESPONSE: We thank the referee for his/her comments, which have helped us improve the manuscript. A detailed point-by-point response to the referee's concerns is provided below.

**Specific comments**

**1. A good portion of the manuscript is devoted to the validation of the model (SWASH) against laboratory data, after which the authors conclude that such a validation justifies application of the model to the field case study. The problem with this line of reasoning should be evident and weakens the paper significantly. The numerical model SWASH has previously been validated (extensively) against laboratory experiments, so this section in itself does not add much to the present study. What one would expect instead is a calibration/validation of the model against field data from Puerto Morelos (the site selected for this research) before carrying out the rest of the study. If such data were not available, the manuscript should probably be reformulated as a more theoretical study and all necessary assumptions (e.g. on bed friction coefficients) should be justified.**

RESPONSE: The section on the laboratory experiments has been removed in accordance with suggestions made by the three referees. Field data for calibrating the model were not available. Therefore justifications of necessary assumptions, including bed friction have been included. The following text has been included in the revised version (page 5, line 20--21):

*"...Although likely to be lower than values obtained in field studies, in the absence of measured values for the study site, this coefficient was used in the numerical simulations. ..."*

**2. The authors confess (e.g. page 9 line 30) that changes in reef roughness are important, but yet have not been considered in this study. Understandably, some assumptions need to be adopted (such as 1D approach, which may miss many important real 2D phenomena, but is a good first approximation), but variable reef roughness for degraded scenarios does not seem to be particularly cumbersome to include in the simulations. Hence, I would recommend that the authors either include variations in reef roughness for different degradations scenarios or justify why this has not been done.**

RESPONSE: The focus of the present study is on the effects of vertical erosion of the reef and dune morphology more than the effects of changes in roughness, which although important, require different scales to be considered and are beyond the scope of this study. Thus, we include the following sentences justifying the assumption (page 5, line 21-25):

*"This study focuses on the degradation of the reef-dune morphology. Reef roughness changes also play an important role in wave transformation (Franklin et al., 2013; Buckley et al., 2016) but high resolution Computational Fluid Dynamics (CFD) modelling is required to allow reef roughness to be taken into account explicitly (e.g. Osorio-Cano et al., sub judice). Therefore the study of these effects is beyond the scope of the present work."*

**3. In line with the previous point, study of the effect of a degraded sand dune, by means of a modified dune height, is an interesting aspect of this study. However, I wonder about the validity of the conclusions achieved regarding flooding (storm impact) when the sand dune has been reduced in height but considered non-erodible during the simulation. A discussion on how this assumption affects the conclusions would be valuable. Ideally, inclusion of morphological evolution of the dune/beach profile in the study of protective services provided by the reef-dune system would significantly strengthen the point made by this article (according to the authors themselves; page 9 line 28).**

RESPONSE: We agree that treating the dune and beach as non-erodible features is a big assumption. Unfortunately, the current model does not have the option for treating the dune or the beach as erodible features. Conducting a hindcast study (1800 simulations) to assess the effects of reef-dune geomorphology and storm surge with a sediment transport model is computationally not feasible for us. Therefore, the following statement is now included in the conclusions of the manuscript (page 14, line 32-page 15, line 4):

*"However, the main drawback in the present study is that it does not consider the dune or the beach as erodible features. Both play an important role in energy dissipation and hence further research is warranted to investigate its effects on increasing/decreasing the storm impact during extreme events. Furthermore, the role of reef roughness and two-dimensional horizontal processes need to be addressed for a more comprehensive study on the implication of reef degradation in such environments."*

Furthermore, the text (page 12, line 13-18) has been modified in the revised manuscript:

*It is also important to note that during an extreme event, such as Hurricane Wilma, the reef can act as a barrier against sediment transport, further reducing the storm impact on the coast by retaining sand in the lagoon and on the beach. However, this is not taken into account in the present study, nor is the effect of changes in reef roughness associated with degradation, which have been shown to have important implications in wave transformation (Buckley et al., 2016) and wave runup (Osorio et al., 2017) but are not the focus of the present study. Furthermore, it is likely that by treating the dune as a non-erodible feature, overtopping is underestimated.*

**4. The paper could be written in a more concise manner by avoiding excess of uninformative or non-relevant details all throughout the manuscript (especially true for Section 2).**

RESPONSE: We agree with the referee and hence most of this information has been removed from this section in the revised manuscript in accordance with suggestions made by all three referees. The following text has been removed:

*"The climate in the region is hot and humid with a mean annual air temperature of 26.4°C, a maximum of 34.5°C in the summer and a minimum of 13°C in the winter (Merino and Otero, 1991). Rainfall is present all year round, although more intense during the summer, with a mean annual rainfall of 1,041 mm (Caribbean Coastal Marine Productivity Program: CARICOMP, unpublished data for the period 1993-1998). Evaporation varies from 102 mm in December to 178 mm in May (Merino and Otero 1983). The mean relative humidity is 84% (CONANP 2000). The water temperature at the bottom of the lagoon varies seasonally by around 5°C, from 31-32°C in August and September, to 24-25°C between December and March (Coronado et al., 2007)."*

*"The lagoon bed is characterised by calcareous sand covered by patches of seagrass with occasional coral colonies."*

*"The mean grain size of the sediment is 0.3 mm, and the mean lower beach slope is 0.05 (Ruiz de Alegría-Arazaburu et al., 2013)."*

**5. No reference is given for the adopted projections of reef erosion (page 7 line 15).**

RESPONSE: This information has been included (page 9, line 12-20). The following text has been added in the manuscript:

*These scenarios were selected based on 50-year projections of reported reef erosion values. For instance, the vertical loss of 6 mm yr$^{-1}$ reported by Sheppard et al. (2005) was used for scenario (ii), whereas the value of 22 mm yr$^{-1}$ reported by Eakin (1996) was used for scenario (iii).*

*The erosion values reported in prior studies are a result of el Niño and bleaching events, which resulted in massive coral mortality and the subsequent erosion of the remaining limestone structure (Sheppard et al. 2005). In recent decades, mass coral bleaching has increased in intensity and frequency (Hoegh-Guldberg et al., 1999), preventing shallow corals from recovering and leading to their gradual disintegration (Sheppard et al., 2005). This is primarily associated with increased temperature, ocean acidification and sea level rise (Hoegh-Guldberg et al., 1999, 2005 and 2007; Pickering et al., 2017). Hence a projection of the above values was used assuming that reefs will continue to erode at similar rates.*

**6. I am not sure all figures are very useful or transmit their message in a clear way. For example, Fig 3 could be transformed into a statistical measure of the goodness of fit between model and experiments. Similarly, Fig 7 is not very informative – the y- axis could probably be presented as the percentage increase/decrease in Ru2% with respect to a reference case (e.g. current profile).**

RESPONSE: We agree with the referee with respect to Fig. 7, and have removed this figure from the revised manuscript. Figure 3 has been removed along with the section on laboratory experiments.

**7. In general, the manuscript is well structured and written, but is not completely free from typos and grammatically confusing sentences. A general revision of the writing is recommended.**

RESPONSE: The manuscript has been reviewed for typos and grammatically confusing sentences.

Furthermore, following the main referees' comments the revised manuscript contains (i) a more thorough analysis on the role that storm surge has in the storm impact, and (ii) a detailed study of runup dynamics.

The reason for not including the storm surge contribution in the previous version of the manuscript is the lack of long tidal records in the study area and that the Hycom data only encompasses 16 years of the 30 years of data corresponding to the wave hindcast information. However, we recognise the importance of studying the role of the storm surge using the available information. Therefore, the numerical model has been re-run selecting 300 representative cases, for the 16-year Hycom period (using the same methodology as for the 30-year hindcast), using: (i) the sea surface height obtained from Hycom (mean sea level including storm surge and astronomical tide) and (ii) the astronomical tide (see Section 5.4 in the revised manuscript - page 11, line 23 to page 12, line 18). The numerical results made it possible compare the effect of including this contribution on the extreme water levels and the storm impact.

The following text has been included in the revised manuscript (page 11, line 23-page 12, line 18):

[revised manuscript text omitted]

**Summary List of Manuscript Edits**

Please find below a summary of the relevant changes made to the manuscript by the authors in response to the above reviews. A fully marked-up version of the manuscript can be found below this summary.

- Section 2 has been modified to remove unnecessary information following the suggestions of all three referees.

- Section 3 has been changed to remove the laboratory experiment from the manuscript in accordance with observations made by the referees. Furthermore this section now includes a justification on the use of the chosen roughness coefficient to address the comments made by Referee#2, and an explanation as to why changes in reef roughness have not been taken into account as suggested by Referee#3.

- Section 4 has been modified to include greater detail on the methodology and analysis used in the present study to address the general comment made by Referee#2. This section also includes references and information for the adopted reef erosion projections, following the observation made by Referee#3.

- Section 5 now includes a more detailed analysis on runup and the effects of reef degradation in order to address observations made by Referee#1 and Referee#2. Furthermore a new subsection (5.4 Role of storm surge) has been included to study the effects of the storm surge on extreme water levels in accordance with comments made by Referee#1 and Referee#2.

- Section 6 has been rewritten as a discussion on the applicability of current runup parameterisations to a reef environment.

- The conclusions have been rewritten to take into account the new results and analyses.

- Extreme runup and wave setup have been redefined as $R_{2\%}$ and $<\eta>$ throughout the revised manuscript.

- Typos have been corrected throughout the text.

- References have been modified in accordance with changes made in the text.

- Figure 1 has been changed to address comments made by Referee#1.

- The original Figure 2 and Figure 3 have been removed along with the section on the laboratory experiment.

- Figure 4 in the original manuscript is now Figure 2.

- Figure 5 in the original manuscript is now Figure 3.

- Figure 4 is a new figure included to provide a better description of the methodology and analysis used in accordance with Referee#2's suggestions.

- Figure 6 has been removed, following the comment made by Referee#2.

- Figure 7 has been removed, following the comment made by Referee#3.

- Figure 8 in the original manuscript is now Figure 5.

- Figure 6 is a new figure which has been added to help explain trends identified in Figure 5 in order to address comments by Referee#2 and Referee#3.

- Figure 7 is a new figure showing the effects of the storm surge in response to observations made by Referee#2 and Referee#3.

- Figure 9 in the original manuscript is now Figure 8 and has been improved.

- Figure 10 in the original manuscript has been removed from the revised version since the effects of the storm surge are now incorporated in the new results and Figure 7.

- Figure 9 and Figure 10 are new figures to show the applicability of current runup parameterizations to a reef environment.

[revised manuscript text omitted]

---

## Referee Report (RR1)

Review of "The role of reef-dune systems in coastal protection in Puerto Morelos (Mexico)" by Franklin et al. This is my second review of this paper. I find the revised version much improved but still needing a few improvements prior to publication.

Specific comments:

- 1. Pg 4, top, "Wave information for a depth..." Please rephrase, as written it is unclear if the model output is written at a depth of 20 m offshore of the study site or at some other location.
- 2. Pg 4 line 7. As before, providing only and R^2 does not describe model skill only how well the model reproduces variance.
- 3. Pg 4 line 9. Provide units for RMS errors and see my above comment re correlation coefficients.
- 4. Section 3, how many layers did you use in SWASH?
- 5. Section 3, description of roughness. Using the same Manning roughness for both the beach and reef is not ideal. You will either over/under-estimate the roughness at the beach/reef. Also the inverse depth dependence of Manning roughness would seem to have a strong impact on the very thin swash lens which accounts for much of your runup. I also don't agree you need CFD to account for roughness, there are quite a few papers that report Cd/roughness lengths for reefs/beaches.
- 6. Pg 6 line 18, I think this should be more than 0.01 not less.
- 7. Pg 6 line 22. Your definitions here are a bit inconsistent. On page 5 Z is defined as mean sea level. In fig 4 Z is the tidal level. Also in SWASH and as illustrated in fig 4 your model results are relative to the tidal level so what you are calling R2 in fig 4 is actually Rhigh as defined in the text. Maybe just rephrase to say you calculate Rhigh from the model and R2 is Rhigh minus the tide.
- 8. Pg 8. Line 6. Should be Rhigh<Dlow
- 9. Page 9, first paragraph. I do not see the shift in the data at 30 m on the x axis. Maybe it would be more clear if you fit regression lines to the data and talked about the slopes. Also you are asking the reader to trust you about the steeper slope of the radiation stress curve and you suddenly bring up reflection from the reef crest. These are not trivial issues, esp reflection from the reef crest, if you are going to say this you need to show something that supports it.
- 10. The storm surge discussion reads as a bit of a bolt on, also I don't understand why you are using the surge and tide or just the tide. Wouldn't the tide only runs be the same as the previous 600 simulations?
- 11. Pg 11, line 7. I haven't read the Stockdon paper in a while but why are you excluding beach slope here?
- 12. Discussion section. I can't say I am a huge fan of the Stockdon comparison, it is not a surprise that a formulation developed on dissipative beaches doesn't work that great on reef but I guess I do see the value in showing it if it prevents one other person from assuming they can use the Stockdon formulation along a reef coast. It would be nice if the discussion section mainly put your results in a broader context. For example, many reef fringed coastlines are not backed by dunes of any significance (e.g. Pacific Islands) such that an increase in runup will be even more detrimental.

---

## Author Response (AR3)

**Response to comments of anonymous referee #1:**
**"The role of the reef-dune system in coastal protection in Puerto Morelos (Mexico)"**

(nhess-2017-304)

by Gemma L. Franklin, Alec Torres-Freyermuth, Gabriela Medellin, María Eugenia Allende-Arandia, Christian M. Appendini

**Referee #1:**
**Review of "The role of reef-dune systems in coastal protection in Puerto Morelos (Mexico)" by Franklin et al. This is my second review of this paper. I find the revised version much improved but still needing a few improvements prior to publication.**

RESPONSE: We thank the referee for his/her valuable comments, which have helped us to further improve the manuscript. A detailed point-by-point response to the referee's comments is provided below.

**Specific comments**

**1. Pg 4, top, "Wave information for a depth..." Please rephrase, as written it is unclear if the model output is written at a depth of 20 m offshore of the study site or at some other location.**

RESPONSE: We have rewritten the sentence for clarity as:

"Wave information is available for a site located at a depth of approximately 20 m offshore of the study site from a 30-year hindcast (1979-2008) for the Gulf of Mexico and the Western Caribbean Sea (Appendini et al., 2014)."

**2. Pg 4 line 7. As before, providing only and R^2 does not describe model skill only how well the model reproduces variance.**

RESPONSE: We thank the referee for his/her comment and understand said concern and hence the reader is referenced to the original paper where the assessment of the wave hindcast was first introduced. Moreover, we have also included the values for bias and scatter in addition to the Rms and correlation coefficients:

"…(Rms of 0.33 m $H_s$ and 1.59 s for $T_p$ with correlation coefficients of 0.90 and 0.51, bias of 0.09 and 0.57, and scatter index of 0.27 and 0.24 respectively). "

**3. Pg 4 line 9. Provide units for RMS errors and see my above comment re correlation coefficients.**

RESPONSE: We thank the referee for noting this and have now included the units (see above).

**4. Section 3, how many layers did you use in SWASH?**

RESPONSE: One layer was used. This was found to be sufficient for the conditions modelled here. We have added this to the following text in the manuscript (pg. 4, line 13-15):

"The Simulating WAves till Shore (SWASH) model, which is a phase-resolving nonlinear non-hydrostatic model (http://swash.sourceforge.net) developed at Delft University of Technology (Zijlema et al., 2011), is used in depth-average mode in this study."

**5. Section 3, description of roughness. Using the same Manning roughness for both the beach and reef is not ideal. You will either over/under-estimate the roughness at the beach/reef. Also the inverse depth dependence of Manning roughness would seem to have a strong impact on the very thin swash lens which accounts for much of your runup. I also don't agree you need CFD to account for roughness, there are quite a few papers that report Cd/roughness lengths for reefs/beaches.**

RESPONSE: We agree that the use of a constant roughness for the profile is not ideal. We have included an additional sentence in the text to acknowledge these restrictions. In order to take into account the 3-dimensional complexity and roughness of reefs, CFD modelling is better, however we recognise that there are studies where Cd/roughness lengths have been reported for reefs and have removed this sentence from the revised manuscript (Pg.4, line 25-29):

"Although likely to be lower than values obtained in field studies, and being a constant value may result in under or over estimation of roughness for the reef or beach respectively, in the absence of measured values for the study site, this coefficient was used in the numerical simulations. Thus, this study focuses on the degradation of the reef-dune morphology. Reef roughness changes also play an important role in wave transformation (Franklin et al., 2013; Buckley et al., 2016). However, the study of these effects is beyond the scope of the present work."

Further work on the effects of roughness on runup is currently underway.

**6. Pg 6 line 18, I think this should be more than 0.01 not less.**

RESPONSE: This sentence should read "less than 0.01" since it is the point used to track the wet-dry boundary. This value was used rather than a value of zero to ensure a continuous time series.

**7. Pg 6 line 22. Your definitions here are a bit inconsistent. On page 5 Z is defined as mean sea level. In fig 4 Z is the tidal level. Also in SWASH and as illustrated in fig 4 your model results are relative to the tidal level so what you are calling R2 in fig 4 is actually Rhigh as defined in the text. Maybe just rephrase to say you calculate Rhigh from the model and R2 is Rhigh minus the tide.**

RESPONSE: We thank the referee for his/her observations. The definition of Z is mean sea level and is the result of astronomical tide plus storm surge. This definition is reduced to tidal level when storm surge is neglected. We have clarified this in the manuscript for consistency. With regards to the definition in Fig. 4, an additional step, which was not previously included in the figure, has been included to explain that tidal level was subtracted in the final step in order to obtain $R_{2\%}$, which is the result of waves (swash+setup). Z was later added to this value to obtain $R_{high}$, as defined in the text ($R_{high}=Z+R_{2\%}$).

**8. Pg 8. Line 6. Should be Rhigh < Dlow.**

RESPONSE: We agree with the referee and have corrected this.

**9. Page 9, first paragraph. I do not see the shift in the data at 30 m on the x axis. Maybe it would be more clear if you fit regression lines to the data and talked about the slopes. Also you are asking the reader to trust you about the steeper slope of the radiation stress curve and you suddenly bring up reflection from the reef crest. These are not trivial issues, esp reflection from the reef crest, if you are going to say this you need to show something that supports it.**

RESPONSE: We have clarified this in the text. The shift can be seen primarily in the figure showing $R_{2\%}$. Prior to this point on the x-axis, the regression slopes are different for all three crest heights. This is shown in the figure below where red dots represent the reef degraded by 1.1 m, green by 0.3 m and black the conserved reef. The regression lines (dark red for the 1.1 m degraded scenario, dark green for the 0.3 m and grey for the conserved) plotted on the data prior to this point (highlighted by the vertical, dashed black line) demonstrate this. Prior to 30 m, the slopes are considerably different for the three profiles, however after this point the slopes are more alike, which suggests that the effect of the reef crest height on this parameter changes around

this point. We have modified the text in the manuscript where this trend is discussed. The following text has been added (page 9, line 3-5):

"This is particularly notable in Figure 6d, as demonstrated by the consistently larger values of $R_{2\%}$ for 1.1 m degraded scenario and $H_0L_0^{\wedge 1/2}$ <30 m, after which there is greater overlap in the values for all three scenarios."

Furthermore, the discussion concerning radiation stress gradient and reflection at the crest has been removed.

[Figure]

**10. The storm surge discussion reads as a bit of a bolt on, also I don't understand why you are using the surge and tide or just the tide. Wouldn't the tide only runs be the same as the previous 600 simulations?**

RESPONSE: We used both for fair comparison in order to see the effects of the storm surge on the results. Since the 300 cases were selected to represent the 16 year time period for which storm surge data were available, and the 600 cases for the 30 year period, the conditions selected are not necessarily the same, hence the 300 tide only scenarios were run to ensure fair comparison.

**11. Pg 11, line 7. I haven't read the Stockdon paper in a while but why are you excluding beach slope here?**

RESPONSE: We used this expression since the work by Stockdon et al. found that the correlation for the parameterisation excluding beach slope was higher than that which included the slope (see equation 12 in Stockdon et al., 2006).

**12. Discussion section. I can't say I am a huge fan of the Stockdon comparison, it is not a surprise that a formulation developed on dissipative beaches doesn't work that great on reef but I guess I do see the value in showing it if it prevents one other person from assuming they can use the Stockdon formulation along a reef coast. It would be nice if the discussion section mainly put your results in a**

**broader context. For example, many reef fringed coastlines are not backed by dunes of any significance (e.g. Pacific Islands) such that an increase in runup will be even more detrimental.**

RESPONSE: We decided to include this comparison in order to highlight such limitations in their application to beaches fringed by reefs and the need for a parameterisation that takes into account other factors such as reef geometry. Furthermore, following the reviewer's suggestion the discussion now puts our results in a broader context, regarding coastlines that are not backed by significant dunes. The following text has been included (pg. 10, line 25 – pg. 11, line 1-2):

[revised manuscript text omitted]